# Policy Optimization with Stochastic Mirror Descent

## Abstract

Improving sample efficiency has been a longstanding goal in reinforcement learning. In this paper, we propose the VRMPO: a sample efficient policy gradient method with stochastic mirror descent. A novel variance reduced policy gradient estimator is the key of VRMPO to improve sample efficiency. Our VRMPO needs only $\mathcal{O}(\epsilon^{-3})$ sample trajectories to achieve an $\epsilon$-approximate first-order stationary point, which matches the best-known sample complexity. We conduct extensive experiments to show our algorithm outperforms state-of-the-art policy gradient methods in various settings.

## 1 Introduction

Reinforcement learning (RL) is one of the most wonderful fields of artificial intelligence, and it has achieved great progress recently (Mnih et al., 2015; Silver et al., 2017). To learn the optimal policy from the delayed reward decision system is the fundamental goal of RL. Policy gradient methods (Williams, 1992; Sutton et al., 2000) are powerful algorithms to learn the optimal policy.

Despite the successes of policy gradient method, suffering from high sample complexity is still a critical challenge for RL. Many existing popular methods require more samples to be collected for each step to update the parameters (Silver et al., 2014; Lillicrap et al., 2016; Schulman et al., 2015; Mnih et al., 2016; Haarnoja et al., 2018), which partially reduces the effectiveness of the sample. Although all the above existing methods claim it improves sample efficiency, they are all empirical results which lack a strong theory analysis of sample complexity.

To improve sample efficiency, in this paper, we explore how to design an efficient and stable algorithm with stochastic mirror descent (SMD). Due to its advantage of the simplicity of implementation, low memory requirement, and low computational complexity (Nemirovsky & Yudin, 1983; Beck & Teboulle, 2003; Lei & Tang, 2018), SMD is one of the most widely used methods in machine learning. However, it is not sound to apply SMD to policy optimization directly, and the challenges are two-fold: **(I)** The objective of policy-based RL is a typical non-convex function, but Ghadimi et al. (2016) show that it may cause instability and even divergence when updating the parameter of a non-convex objective function by SMD via a single batch sample. **(II)** Besides, the large variance of gradient estimator is the other bottleneck of applying SMD to policy optimization for improving sample efficiency. In fact, in reinforcement learning, the non-stationary sampling process with the environment leads to the large variance of existing methods on the estimate of policy gradient, which results in poor sample efficiency (Papini et al., 2018; Liu et al., 2018).

**Contributions** To address the above two problems correspondingly, in this paper **(I)** We analyze the theoretical dilemma of applying SMD to policy optimization. Our analysis shows that under the common Assumption 1, for policy-based RL, designing the algorithm via SMD directly can not guarantee the convergence. Hence, we propose the MPO algorithm with a provable convergence guarantee. Designing an efficiently computable, and unbiased gradient estimator by averaging its historical policy gradient is the key to MPO. **(II)** We propose the VRMPO: a sample efficient policy optimization algorithm via constructing a variance reduced policy gradient estimator. Specifically, we propose an efficiently computable policy gradient estimator, utilizing fresh information and yielding a more accurate estimation of the gradient w.r.t the objective, which is the key to improve sample efficiency. We prove VRMPO needs $\mathcal{O}(\epsilon^{-3})$ sample trajectories to achieve an $\epsilon$-approximate first-order stationary point ($\epsilon$-FOSP) (Nesterov, 2004). To our best knowledge, our VRMPO matches the

best-known sample complexity among the existing literature. Besides, we conduct extensive experiments, which further show that our algorithm outperforms state-of-the-art bandit algorithms in various settings.

## 2 BACKGROUND AND NOTATIONS

### 2.1 POLICY-BASED REINFORCEMENT LEARNING

We consider the *Markov decision processes* $M = (\mathcal{S}, \mathcal{A}, \mathcal{P}, \mathcal{R}, \rho_0, \gamma)$, where $\mathcal{S}$ is state space, $\mathcal{A}$ is action space; At time $t$, the agent is in a state $S_t \in \mathcal{S}$ and takes an action $A_t \in \mathcal{A}$, then it receives a feedback $R_{t+1}$; $P_{ss'}^a = P(s'|s,a) \in \mathcal{P}$ is the probability of the state transition from $s$ to $s'$ under taking $a \in \mathcal{A}$; The bounded reward function $\mathcal{R} : \mathcal{S} \times \mathcal{A} \to [-R, R]$, $\mathcal{R}_s^a \mapsto \mathbb{E}[R_{t+1}|S_t = s, A_t = a]$; $\rho_0 : \mathcal{S} \to [0,1]$ is the initial state distribution and $\gamma \in (0,1)$ is discounted factor. Policy $\pi_\theta(a|s)$ is a probability distribution on $\mathcal{S} \times \mathcal{A}$ with the parameter $\theta \in \mathbb{R}^p$. Let $\tau = \{s_t, a_t, r_{t+1}\}_{t=0}^{H_\tau}$ be a trajectory, where $s_0 \sim \rho_0(s_0)$, $a_t \sim \pi_\theta(\cdot|s_t)$, $r_{t+1} = \mathcal{R}(s_t, a_t)$, $s_{t+1} \sim P(\cdot|s_t, a_t)$, and $H_\tau$ is the finite horizon of $\tau$. The expected return $J(\pi_\theta)$ is defined as:

$$J(\theta) \stackrel{\text{def}}{=} J(\pi_\theta) = \int_\tau P(\tau|\theta)R(\tau)\mathrm{d}\tau = \mathbb{E}_{\tau \sim \pi_\theta}[R(\tau)], \tag{1}$$

where $P(\tau|\theta) = \rho_0(s_0)\prod_{t=0}^{H_\tau} P(s_{t+1}|s_t, a_t)\pi_\theta(a_t|s_t)$ is the probability of generating $\tau$, $R(\tau) = \sum_{t=0}^{H_\tau} \gamma^t r_{t+1}$ is the accumulated discounted return.

Let $\mathcal{J}(\theta) = -J(\theta)$, the central problem of policy-based RL is to solve the problem:

$$\theta^* = \arg\max_\theta J(\theta) \iff \theta^* = \arg\min_\theta \mathcal{J}(\theta). \tag{2}$$

Computing the $\nabla J(\theta)$ analytically, we have

$$\nabla J(\theta) = \mathbb{E}_{\tau \sim \pi_\theta}[\sum_{t=0}^{H_\tau} \nabla_\theta \log \pi_\theta(a_t|s_t)R(\tau)]. \tag{3}$$

For any trajectory $\tau$, let $g(\tau|\theta) = \sum_{t=0}^{H_\tau} \nabla_\theta \log \pi_\theta(a_t|s_t)R(\tau)$, which is an unbiased estimator of $\nabla J(\theta)$. Vanilla policy gradient (VPG) is a straightforward way to solve problem (2):

$$\theta \leftarrow \theta + \alpha g(\tau|\theta),$$

where $\alpha$ is step size.

**Assumption 1.** (Sutton et al., 2000; Papini et al., 2018) *For each pair $(s,a)$, any $\theta \in \mathbb{R}^p$, and all components $i$, $j$, there exists positive constants $G$, $F$ s.t.,*

$$|\nabla_{\theta_i} \log \pi_\theta(a|s)| \le G, \quad |\frac{\partial^2}{\partial\theta_i\partial\theta_j} \log \pi_\theta(a|s)| \le F. \tag{4}$$

According to the Lemma B.2 of (Papini et al., 2018), Assumption 1 implies $\nabla J(\theta)$ is $L$-Lipschiz, i.e., $\|\nabla J(\theta_1) - \nabla J(\theta_2)\| \le L\|\theta_1 - \theta_2\|$, where

$$L = RH(HG^2 + F)/(1 - \gamma), \tag{5}$$

Besides, Assumption 1 implies the following property of the policy gradient estimator.

**Lemma 1** (Properties of stochastic differential estimators (Shen et al., 2019)). *Under Assumption 1, for any policy $\pi_\theta$ and $\tau \sim \pi_\theta$, we have*

$$\|g(\tau|\theta) - \nabla J(\theta)\|^2 \le \frac{G^2 R^2}{(1-\gamma)^4} \stackrel{\text{def}}{=} \sigma^2. \tag{6}$$

### 2.2 STOCHASTIC MIRROR DESCENT

Now, we review some basic concepts of SMD; in this section, the notation follows (Nemirovski et al., 2009). Let's consider the stochastic optimization problem,

$$\min_{\theta \in D_\theta}\{f(\theta) = \mathbb{E}[F(\theta; \xi)]\}, \tag{7}$$

where $D_\theta \in \mathbb{R}^n$ is a nonempty convex compact set, $\xi$ is a random vector whose probability distribution, $\mu$ is supported on $\Xi \in \mathbb{R}^d$ and $F : D_\theta \times \Xi \to \mathbb{R}$. We assume that the expectation $\mathbb{E}[F(\theta; \xi)] = \int_\Xi F(\theta; \xi) \mathrm{d}\mu(\xi)$ is well defined and finite-valued for every $\theta \in D_\theta$.

**Definition 1** (Proximal Operator (Moreau, 1965)). *$T$ is a function defined on a closed convex $\mathcal{X}$, and $\alpha > 0$. $\mathcal{M}_{\alpha,T}^\psi(z)$ is the proximal operator of $T$, which is defined as:*

$$\mathcal{M}_{\alpha,T}^\psi(z) = \arg\min_{x \in \mathcal{X}} \{T(x) + \frac{1}{\alpha} D_\psi(x, z)\}, \tag{8}$$

*where $\psi(x)$ is a continuously-differentiable, $\zeta$-strictly convex function: $\langle x - y, \nabla\psi(x) - \nabla\psi(y) \rangle \geq \zeta \|x - y\|^2$, $\zeta > 0$, $D_\psi$ is Bregman distance: $D_\psi(x, y) = \psi(x) - \psi(y) - \langle \nabla\psi(y), x - y \rangle$, $\forall\, x, y \in \mathcal{X}$.*

**Stochastic Mirror Descent** The SMD solves (7) by generating an iterative solution as follows,

$$\theta_{t+1} = \mathcal{M}_{\alpha_t, \ell(\theta)}^\psi(\theta_t) = \arg\min_{\theta \in D_\theta} \{\langle g_t, \theta \rangle + \frac{1}{\alpha_t} D_\psi(\theta, \theta_t)\}, \tag{9}$$

where $\alpha_t > 0$ is step-size, $\ell(\theta) = \langle g_t, \theta \rangle$ is the first-order approximation of $f(\theta)$ at $\theta_t$, $g_t = g(\theta_t, \xi_t)$ is stochastic subgradient such that $g(\theta_t) = \mathbb{E}[g(\theta_t, \xi_t)] \in \partial f(\theta)|_{\theta=\theta_t}$, $\{\xi_t\}_{t \geq 0}$ represents a draw form distribution $\mu$, and $\partial f(\theta) = \{g | f(\theta) - f(\omega) \leq g^T(\theta - \omega), \forall \omega \in \mathbf{dom}(f)\}$.

If we choose $\psi(x) = \frac{1}{2}\|x\|_2^2$, which implies $D_\psi(x, y) = \frac{1}{2}\|x - y\|_2^2$, since then iteration (9) is the proximal gradient (Rockafellar, 1976) view of SGD. Thus, SMD is a generalization of SGD.

**Convergence Criteria: Bregman Gradient** Bregman gradient is a generation of projected gradient (Ghadimi et al., 2016). Recently, Zhang & He (2018); Davis & Grimmer (2019) develop it to measure the convergence of an algorithm for the non-convex optimization problem. Evaluating the difference between each candidate solution $x$ and its proximity is the critical idea of Bregman gradient to measure the stationarity of $x$. Specifically, let $\mathcal{X}$ be a closed convex set on $\mathbb{R}^n$, $\alpha > 0$, $T(x)$ is defined on $\mathcal{X}$. The Bregman gradient of $T$ at $x \in \mathcal{X}$ is:

$$\mathcal{G}_{\alpha,T}^\psi(x) = \alpha^{-1}(x - \mathcal{M}_{\alpha,T}^\psi(x)), \tag{10}$$

where $\mathcal{M}_{\alpha,T}^\psi(\cdot)$ is defined in Eq.(8). If $\psi(x) = \frac{1}{2}\|x\|_2^2$, then $x^*$ is a critical point of $T$ if and only if $\mathcal{G}_{\alpha,T}^\psi(x^*) = \nabla T(x^*) = 0$ (Bauschke et al. (2011);Theorem 27.1). Thus, Bregman gradient (10) is a generalization of gradient. The following Remark 1 is helpful for us to understand the significance of Bregman gradient, and it gives us some insights to understand this convergence criterion.

**Remark 1.** *Let $T$ be a convex function, by the Proposition 5.4.7 of Bertsekas (2009): $x^*$ is a stationarity point of $T$ if and only if*

$$0 \in \partial(T + \delta_\mathcal{X})(x^*), \tag{11}$$

*where $\delta_\mathcal{X}(\cdot)$ is the indicator function on $\mathcal{X}$. Furthermore, suppose $\psi(x)$ is twice continuously differentiable, let $\tilde{x} = \mathcal{M}_{\alpha,T}^\psi(x)$, by the definition of proximal operator $\mathcal{M}_{\alpha,T}^\psi(\cdot)$, we have*

$$0 \in \partial(T + \delta_\mathcal{X})(\tilde{x}) + \left(\nabla\psi(\tilde{x}) - \nabla\psi(x)\right) \overset{(\star)}{\approx} \partial(T + \delta_\mathcal{X})(\tilde{x}) + \alpha\mathcal{G}_{\alpha,T}^\psi(x)\nabla^2\psi(x), \tag{12}$$

*Eq.($\star$) holds due to the first-order Taylor expansion of $\nabla\psi(x)$. By the criteria of (11), if $\mathcal{G}_{\alpha,T}^\psi(x) \approx 0$, Eq.(12) implies the origin point $0$ is near the set $\partial(T + \delta_\mathcal{X})(\tilde{x})$, i.e., $\tilde{x}$ is close to a stationary point.*

In practice, we choose $T(\theta) = \langle -\nabla J(\theta_t), \theta \rangle$, since then discriminant criterion (12) is suitable to RL problem (2). For the non-convex problem (2), we are satisfied with finding an $\epsilon$-approximate First-Order Stationary Point ($\epsilon$-FOSP) (Nesterov, 2004), denoted by $\theta_\epsilon$, such that

$$\|\mathcal{G}_{\alpha,T(\theta_\epsilon)}^\psi(\theta_\epsilon)\| \leq \epsilon. \tag{13}$$

## 3 POLICY OPTIMIZATION WITH STOCHASTIC MIRROR DESCENT

In this section, we solve the problem (2) via SMD. Firstly, we analyze the theoretical dilemma of applying SMD directly to policy optimization. Then, we propose a convergent mirror policy optimization algorithm (MPO).

### 3.1 THEORETICAL DILEMMA

Let $\mathcal{T} = \{\tau_k\}_{k=0}^{N-1}$ be a collection of trajectories, $\tau_k \sim \pi_{\theta_k}$, we receive gradient information:

$$-g(\tau_k|\theta_k) = -\sum_{t=0}^{H_{\tau_k}} \nabla_\theta \log \pi_\theta(a_t|s_t) R(\tau_k)|_{\theta=\theta_k}, \tag{14}$$

then by SMD (9), to solve (2), for each $0 \le k \le N-1$, we define the update rule as follows,

$$\theta_{k+1} = \mathcal{M}_{\alpha_k, \langle -g(\tau_k|\theta_k), \theta \rangle}^\psi (\theta_k) = \arg\min_\theta \{ \langle -g(\tau_k|\theta_k), \theta \rangle + \frac{1}{\alpha_k} D_\psi(\theta, \theta_k) \}, \tag{15}$$

where $\alpha_k > 0$ is step-size and other symbols are consistent to previous paragraphs. Due to $-J(\theta)$ is non-convex, according to (Ghadimi et al., 2016), a standard strategy for analyzing non-convex optimization methods is to pick up the output $\tilde{\theta}_N$ randomly according to the following distribution over $\{1, 2, \cdots, N\}$:

$$P(\tilde{\theta}_N = \theta_n) = \frac{\zeta\alpha_n - L\alpha_n^2}{\sum_{k=1}^N (\zeta\alpha_k - L\alpha_k^2)}. \tag{16}$$

where $\zeta$ is defined in Definition 1, $0 < \alpha_n < \frac{\zeta}{L}$, $n = 1, 2, \cdots, N$.

**Theorem 1.** (Ghadimi et al., 2016) *Under Assumption 1, and the total trajectories are $\{\tau_k\}_{k=1}^N$. Consider the sequence $\{\theta_k\}_{k=1}^N$ generated by (15), the output $\tilde{\theta}_N = \theta_n$ follows the probability mass distribution of (16). Let $0 < \alpha_k < \frac{\zeta}{L}$, $\ell(g, u) = \langle g, u \rangle$, the term $L$ and $\sigma$ are defined in Eq.(5) and Eq.(6) correspondingly. Then, we have*

$$\mathbb{E}[\|\mathcal{G}_{\alpha_n, \ell(-g_n, \theta_n)}^\psi(\theta_n)\|^2] \le \frac{\left(J(\theta^*) - J(\theta_1)\right) + \frac{\sigma^2}{\zeta} \sum_{k=1}^N \alpha_k}{\sum_{k=1}^N (\zeta\alpha_k - L\alpha_k^2)}, \tag{17}$$

*where $g_n$ is short for $g(\tau_n|\theta_n)$.*

Unfortunately, it is worth to notice that the lower bound of (17) reaches

$$\frac{\left(J(\theta^*) - J(\theta_1)\right) + \frac{\sigma^2}{\zeta} \sum_{k=1}^N \alpha_k}{\sum_{k=1}^N (\zeta\alpha_k - L\alpha_k^2)} \ge \frac{\sigma^2}{\zeta^2}, \tag{18}$$

which can not guarantee the convergence of the iteration (15), no matter how the step-size $\alpha_k$ is specified. Thus, under the Assumption 1, generating the solution $\{\theta_k\}_{k=1}^N$ according to (15) and the output following (16) lack a strong convergence guarantee.

**An Open Problem** The iteration (15) is a very important and general scheme that unifies many existing algorithms. For example, if the mirror map $\psi(\theta) = \frac{1}{2}\|\theta\|_2^2$, then the update (15) is reduced to policy gradient algorithm (Sutton et al., 2000) which is widely used in modern RL. The update (15) is *natural policy gradient* (Kakade, 2002; Peters & Schaal, 2008; Thomas et al., 2013) if we choose mirror map $\psi(\theta) = \frac{1}{2}\theta^\top F(\theta)\theta$, where $F = \mathbb{E}_{\tau \sim \pi_\theta}[\nabla_\theta \log \pi_\theta(s, a) \nabla_\theta \log \pi_\theta(s, a)^\top]$ is Fisher information matrix. If $\psi$ is Boltzmann-Shannon entropy function (Shannon, 1948), then $D_\psi$ is known as KL divergence and update (15) is reduced to relative entropy policy search (Peters et al., 2010; Fox et al., 2016; Chow et al., 2018). Despite the vast body of work around above specific methods, current works are scattered and fragmented in both theoretical and empirical aspects (Agarwal et al., 2019). Thus, it is of great significance to establish the fundamental theoretical convergence properties of iteration (15).

Please notice that for the non-convexity of problem (2), the lower bound of (18) holds under Assumption 1. It is natural to ask:

**What conditions guarantee the convergence of scheme (15)?**

This is an open problem. Although, the iteration (15) is intuitively a convergent scheme, as discussed above that particular mirror maps $\psi$ can lead (15) to some popular empirically effective RL algorithms; there is still no generally complete theoretical convergence analysis of (15). Such convergence properties not only help us to understand better why those methods work but also inspire us to design novel algorithms with the principled approaches. We leave this open problem and the related questions, e.g., how fast the iteration (15) converges to global optimality or its finite sample analysis, as future works.

---

**Algorithm 1** Mirror Policy Optimization Algorithm (MPO)

1: **Initialize:** parameter $\theta_1$, step-size $\alpha_k > 0$, $g_0 = 0$, parametric policy $\pi_\theta(a|s)$, and map $\psi$.
2: **for** $k = 1$ **to** $N$ **do**
3:    Generate a trajectory $\tau_k = \{s_t, a_t, r_{t+1}\}_{t=0}^{H_{\tau_k}} \sim \pi_{\theta_k}$, temporary variable $g_0 = 0$.

$$g_k \leftarrow \sum_{t=0}^{H_{\tau_k}} \nabla_\theta \log \pi_\theta(a_t|s_t) R(\tau_k)|_{\theta=\theta_k} \tag{21}$$

$$\hat{g}_k \leftarrow \frac{1}{k} g_k + (1 - \frac{1}{k}) g_{k-1} \tag{22}$$

$$\theta_{k+1} \leftarrow \arg\min_\omega \{\langle -\hat{g}_k, \omega \rangle + \frac{1}{\alpha_k} D_\psi(\omega, \theta_k)\} \tag{23}$$

4: **end for**
5: **Output:** $\tilde{\theta}_N$ according to (16).

---

### 3.2 An Implementation with Convergent Guarantee

In this section, we propose a convergent implementation defined as follows, for each step $k$:

$$\theta_{k+1} = \mathcal{M}_{\alpha_k, \langle -\hat{g}_k, \theta \rangle}^\psi(\theta_k) = \arg\min_{\theta \in \Theta}\{\langle -\hat{g}_k, \theta \rangle + \frac{1}{\alpha_k} D_\psi(\theta, \theta_k)\}, \tag{19}$$

where $\hat{g}_k$ is an arithmetic mean of previous episodes' gradient estimate $\{g(\tau_i|\theta_i)\}_{i=1}^k$:

$$\hat{g}_k = \frac{1}{k} \sum_{i=1}^k g(\tau_i|\theta_i). \tag{20}$$

We present the details of an implementation in Algorithm 1. Notice that Eq.(22) is an incremental implementation of the average (20), thus, (22) enjoys a lower storage cost than (20).

For a given episode, the gradient flow (20)/(22) of MPO is slightly different from the traditional VPG, REINFORCE (Williams, 1992), A2C (Mnih et al., 2016) or DPG (Silver et al., 2014) whose gradient estimator follows (14) that is according to the trajectory of current episode, while our MPO uses an arithmetic mean of previous episodes' gradients. The estimator (14) is a natural way to estimate the term $-\nabla J(\theta_t) = -\mathbb{E}[\sum_{k=0}^{H_{\tau_t}} \nabla_\theta \log \pi_\theta(a_k|s_k) R(\tau_t)]$, i.e. using a single current trajectory to estimate policy gradient. Unfortunately, under Assumption 1, the result of (18) shows using (14) with SMD lacks a guarantee of convergence. This is exactly the reason why we abandon the way (14) and turn to propose (20)/(22) to estimate policy gradient. We provide the convergence analysis of our scheme (20)/(22) in the next Theorem 2.

**Theorem 2** (Convergence Rate of Algorithm 1). *Under Assumption 1, and the total trajectories are $\{\tau_k\}_{k=1}^N$. Consider the sequence $\{\theta_k\}_{k=1}^N$ generated by Algorithm 1, and the output $\tilde{\theta}_N = \theta_n$ follows the distribution of Eq.(16). Let $0 < \alpha_k < \frac{\zeta}{L}$, $\ell(g, u) = \langle g, u \rangle$, the term $L$ and $\sigma$ are defined in Eq.(5) and Eq.(6) correspondingly. Let $\hat{g}_k = \frac{1}{k} \sum_{i=1}^k g_i$, where $g_i = \sum_{t=0}^{H_{\tau_i}} \nabla_\theta \log \pi_\theta(a_t|s_t) R(\tau_i)|_{\theta=\theta_i}$. Then we have*

$$\mathbb{E}[\|\mathcal{G}_{\alpha_n, \ell(-g_n, \theta_n)}^\psi(\theta_n)\|^2] \le \frac{\left(J(\theta^*) - J(\theta_1)\right) + \frac{\sigma^2}{\zeta} \sum_{k=1}^N \frac{\alpha_k}{k}}{\sum_{k=1}^N (\zeta \alpha_k - L \alpha_k^2)}. \tag{24}$$

We prove the proof in Appendix A. Let $\alpha_k = \zeta/2L$, then, Eq(24) is $\mathbb{E}[\|\mathcal{G}_{\alpha_n, \ell(-\hat{g}_n, \theta_n)}^\psi(\theta_n)\|^2] \le \frac{4L\left(J(\theta^*) - J(\theta_1)\right) + 2\sigma^2 \sum_{k=1}^N \frac{1}{k}}{N\zeta^2} = \mathcal{O}(\ln N/N)$. Our scheme of MPO partially answers the previous open problem through conducting a new policy gradient estimator.

## 4 VRMPO: A Variance Reduction Implementation of MPO

In this section, we propose a variance reduction version of MPO: VRMPO. In optimization community, *variance reduction gradient estimator* is a very popular method with provable convergence

---

**Algorithm 2** Variance-Reduced Mirror Policy Optimization (VRMPO).

1: **Initialize:** Policy $\pi_\theta(a|s)$ with parameter $\tilde{\theta}_0$, mirror map $\psi$, step-size $\alpha_k > 0$, epoch size $K, m$.
2: **for** $k = 1$ **to** $K$ **do**
3:   $\theta_{k,0} = \tilde{\theta}_{k-1}$, generate $\mathcal{T}_k = \{\tau_i\}_{i=1}^{N_1} \sim \pi_{\theta_{k,0}}$
4:   $\theta_{k,1} = \theta_{k,0} - \alpha_k G_{k,0}$, where $G_{k,0} = -\hat{\nabla}_{N_1} J(\theta_{k,0}) = -\frac{1}{N_1} \sum_{i=1}^{N_1} g(\tau_i|\theta_{k,0})$.
5:   **for** $t = 1$ **to** $m - 1$ **do**
6:     Generate $\{\tau_j\}_{j=1}^{N_2} \sim \pi_{\theta_{k,t}}$

$$G_{k,t} = G_{k,t-1} + \frac{1}{N_2} \sum_{j=1}^{N_2} (-g(\tau_j|\theta_{k,t}) + g(\tau_j|\theta_{k,t-1})), \tag{25}$$

$$\theta_{k,t+1} = \arg\min_\omega \{\langle G_{k,t}, \omega \rangle + \frac{1}{\alpha_k} D_\psi(\omega, \theta_{k,t})\} \tag{26}$$

7:   **end for**
8:   $\tilde{\theta}_k = \theta_{k,t}$ with $t$ chosen uniformly randomly from $\{0, 1, ..., m\}$.
9: **end for**
10: **Output:** $\tilde{\theta}_K$.

---

guarantee (Reddi et al., 2016; Fang et al., 2018; Horváth & Richtárik, 2019). Inspired by the above works, now, we present an efficiently computable policy gradient estimator. For any initial $\theta_0$, let $\{\tau_j^0\}_{j=1}^N \sim \pi_{\theta_0}$, we calculate the initial gradient estimate as follows,

$$G_0 = -\hat{\nabla}_N J(\theta_0) \stackrel{\text{def}}{=} -\frac{1}{N} \sum_{j=1}^N g(\tau_j^0|\theta_0). \tag{27}$$

Let $\theta_1 = \theta_0 - \alpha G_0$, for each time $t \in \mathbb{N}^+$, let $\{\tau_j^t\}_{j=1}^N$ be the trajectories generated by $\pi_{\theta_t}$, we define the policy gradient estimate $G_t$ and the update rule of parameter as follows,

$$G_t = G_{t-1} + \frac{1}{N} \sum_{j=1}^N (-g(\tau_j^t|\theta_t) + g(\tau_j^t|\theta_{t-1})), \tag{28}$$

$$\theta_{t+1} = \arg\min_\theta \{\langle G_t, \theta \rangle + \frac{1}{\alpha} D_\psi(\theta, \theta_t)\}, \tag{29}$$

where $\alpha > 0$ is step-size. We present more details in Algorithm 2.

In (28), $-g(\tau_j^t|\theta_t)$ and $g(\tau_j^t|\theta_{t-1})$ share the same trajectory $\{\tau_j^t\}_{j=1}^N$, which plays a critical role in reducing the variance of gradient estimate (Shen et al., 2019). Besides, it is different from (20), we admit a simple recursive formulation to conduct the policy gradient estimator $G_t$ (28), which captures some techniques from SARAH (Nguyen et al., 2017a;b). At each time $t$, the term $\frac{1}{N} \sum_{j=1}^N \left(-g(\tau_j^t|\theta_t) + g(\tau_j^t|\theta_{t-1})\right)$ can be seen as an additional "noise" for the policy gradient estimate. A lot of practices show that conducting a gradient estimator with the additional noise enjoys a lower variance and speeding up the convergence (Reddi et al., 2016; Schmidt et al., 2017; Nguyen et al., 2017a;b; Fang et al., 2018).

**Theorem 3** (Convergence Analysis of Algorithm 2). *The sequence $\{\tilde{\theta}_k\}_{k=1}^K$ is generated according to Algorithm 2. Under Assumption 1, let $\zeta > \frac{5}{32}$, for any positive scalar $\epsilon$, the batch size of the trajectories of outer loop $N_1 = \left(\frac{1}{8L\zeta^2} + \frac{1}{2(\zeta - \frac{5}{32})}\left(1 + \frac{1}{32\zeta^2}\right)\right)\frac{\sigma^2}{\epsilon^2}$, the iteration times of inner loop $m - 1 = N_2 = \sqrt{\left(\frac{1}{8L\zeta^2} + \frac{1}{2(\zeta - \frac{5}{32})}\left(1 + \frac{1}{32\zeta^2}\right)\right)}\frac{\sigma}{\epsilon}$, the iteration times of outer loop $K = \frac{8L(\mathbb{E}[\mathcal{J}(\tilde{\theta}_0)] - \mathcal{J}(\theta^*))}{(m-1)(\zeta - \frac{5}{32})}\left(1 + \frac{1}{16\zeta^2}\right)\frac{1}{\epsilon^2}$, and step size $\alpha_k = \frac{1}{4L}$. Then, Algorithm 2 outputs the point $\tilde{\theta}_K$ achieves*

$$\mathbb{E}\left[\|\mathcal{G}_{\alpha, \langle -\nabla J(\tilde{\theta}_K), \theta \rangle}^\psi(\tilde{\theta}_K)\|\right] \leq \epsilon. \tag{30}$$

By the result in Theorem 3, under Assumption 1, to achieve the $\epsilon$-FOSP, Algorithm 2 (VRMPO) needs $K(N_1 + (m-1)N_2) = \frac{8L(\mathbb{E}[\mathcal{J}(\tilde{\theta}_0)] - \mathcal{J}(\theta^*))}{(\zeta - \frac{5}{32})} \left(1 + \frac{1}{16\zeta^2}\right) \left(1 + \sqrt{\left(\frac{1}{8L\zeta^2} + \frac{1}{2(\zeta - \frac{5}{32})} \left(1 + \frac{1}{32\zeta^2}\right)\right)} \frac{\sigma}{\epsilon}\right) \frac{1}{\epsilon^2} = \mathcal{O}(\frac{1}{\epsilon^3})$ random trajectories. As far as we know, our VRMPO matches the best-known sample complexity as the HAPG algorithm (Shen et al., 2019).

In fact, according to (Shen et al., 2019), REINFORCE needs $\mathcal{O}(\epsilon^{-4})$ random trajectory trajectories to achieve the $\epsilon$-FOSP, and no provable improvement on its complexity has been made so far. The same order of sample complexity of REINFORCE is shown by Xu et al. (2019). With the additional assumptions $\mathbb{V}\text{ar}\left[\prod_{h=0}^{H} \frac{\pi_{\theta_0}(a_h|s_h)}{\pi_{\theta_t}(a_h|s_h)}\right], \mathbb{V}\text{ar}[g(\tau|\theta)] < +\infty$, Papini et al. (2018) show that the SVRPG achieves the sample complexity of $\mathcal{O}(\epsilon^{-4})$. Later, under the same assumption as (Papini et al., 2018), Xu et al. (2019) reduce the sample complexity of SVRPG to $\mathcal{O}(\epsilon^{-\frac{10}{3}})$. We provide more details of the comparison in Table 1, from which it is easy to see that our VRMPO matches the best-known sample complexity with least conditions.

## 5 RELATED WORKS

**Stochastic Variance Reduced Gradient in RL** Although it has achieved considerable successes in supervised learning, stochastic variance reduced gradient optimization is rarely a matter of choice in RL. To our best knowledge, Du et al. (2017) firstly introduce SVRG (Johnson & Zhang, 2013) to off-policy evaluation. Du et al. (2017) transform the empirical policy evaluation problem into a (quadratic) convex-concave saddle-point problem, then they solve the problem via SVRG straightforwardly. Later, to improve sample efficiency for complex RL, Xu et al. (2017) combine SVRG with TRPO (Schulman et al., 2015). Similarly, Yuan et al. (2019) introduce SARAH (Nguyen et al., 2017a) to TRPO to improve sample efficiency. However, the results presented by Xu et al. (2017) and Yuan et al. (2019) are empirical, which lacks a strong theory analysis. Metelli et al. (2018) present a surrogate objective function with a Rényi divergence (Rényi et al., 1961) to reduce the variance caused by importance sampling.

Recently, Papini et al. (2018) propose a stochastic variance reduced version of policy gradient (SVRPG), and they define the gradient estimator via important sampling,

$$G_t = \widetilde{G}_{t-1} + \frac{1}{N} \sum_{j=1}^{N} \left( -g(\tau_j^t|\theta_t) + \prod_{h=0}^{H} \frac{\pi_{\theta_0}(a_h|s_h)}{\pi_{\theta_t}(a_h|s_h)} g(\tau_j^t|\theta_{t-1}) \right), \quad (31)$$

where $\widetilde{G}_{t-1}$ is an unbiased estimator according to the trajectory generated by $\pi_{\theta_{t-1}}$. Although the above algorithms are practical empirically, their gradient estimates are dependent heavily on important sampling. This fact partially reduces the effectiveness of variance reduction. Later, Shen et al. (2019) remove the important sampling term, and they construct the gradient estimator as follows,

$$G_t = \widetilde{G}_{t-1} + \frac{1}{N} \sum_{j=1}^{N} \left( -g(\tau_j^t|\theta_t) + g(\tau_j^t|\theta_{t-1}) \right). \quad (32)$$

It is different from (Du et al., 2017; Xu et al., 2017; Papini et al., 2018; Shen et al., 2019), the proposed VRMPO admits a stochastic recursive iteration to estimate the policy gradient, see Eq.(28). Our VRMPO exploits the fresh information to improve convergence and reduce variance. Besides, VRMPO reduces the storage cost significantly due to it doesn't require to store the complete historical information. We provide more details of the comparison in Table 1.

**Baseline Methods for Variance Reduction of Policy Gradient** Baseline (also also known as control variates (Cheng et al., 2019a) or reward reshaping (Ng et al., 1999; Jie & Abbeel, 2010)) is a widely used technique to reduce the variance (Weaver & Tao, 2001; Greensmith et al., 2004). For example, A2C (Sutton & Barto, 1998; Mnih et al., 2016) introduces the value function as baseline function, Wu et al. (2018) consider action-dependent baseline, and Liu et al. (2018) use the Stein's identity (Stein, 1986) as baseline. Q-Prop (Gu et al., 2017) makes use of both the linear dependent baseline and GAE (Schulman et al., 2016) to reduce variance. Cheng et al. (2019b) present a predictor-corrector framework that can transform a first-order model-free algorithm into a new hybrid method that leverages predictive models to accelerate policy learning. Mao et al. (2019) derive a bias-free,

| Algorithm | Estimator | Conditions | Complexity |
|---|---|---|---|
| VPG/REINFORCE | Eq.(14) | Assumption 1;$\mathbb{V}\mathrm{ar}[g(\tau|\theta)] < +\infty$ | $\mathcal{O}(\epsilon^{-4})$ |
| SVRPG (Papini et al., 2018) | Eq.(31) | Assumption 1;$\mathbb{V}\mathrm{ar}[\rho_t],\mathbb{V}\mathrm{ar}[g(\tau|\theta)] < +\infty$ | $\mathcal{O}(\epsilon^{-4})$ |
| SVRPG (Xu et al., 2019) | Eq.(31) | Assumption 1;$\mathbb{V}\mathrm{ar}[\rho_t],\mathbb{V}\mathrm{ar}[g(\tau|\theta)] < +\infty$ | $\mathcal{O}(\epsilon^{-10/3})$ |
| HAPG (Shen et al., 2019) | Eq.(32) | Assumption 1 | $\mathcal{O}(\epsilon^{-3})$ |
| VRMPO (Our Works) | Eq.(28) | Assumption 1 | $\mathcal{O}(\epsilon^{-3})$ |

Table 1: Comparison on complexity required to achieve $\|\nabla J(\theta)\| \leq \epsilon$. Particularly, if $\psi(\theta) = \frac{1}{2}\|\theta\|_2^2$, then the result (30) of our VRMPO is measured by gradient. Beside, $\rho_t \stackrel{\text{def}}{=} \prod_{h=0}^{H} \frac{\pi_{\theta_0}(a_h|s_h)}{\pi_{\theta_t}(a_h|s_h)}$.

input-dependent baseline to reduce variance, and analytically show its benefits over state-dependent baselines. Recently, Grathwohl et al. (2018); Cheng et al. (2019a) provide a standard explanation for the benefits of such approaches with baseline function.

However, the capacity of all the above methods is limited by their choice of baseline function (Liu et al., 2018). In practice, it is troublesome to design a proper baseline function to reduce the variance of policy gradient estimate. Our VRMPO avoids the selection of baseline function, and it uses a current sample trajectory to construct a novel, efficiently computable gradient estimator to reduce variance and speed convergence.

## 6 EXPERIMENTS

### 6.1 NUMERICAL ANALYSIS OF MPO

In this section, we use an experiment to demonstrate MPO converges faster than VPG/REINFORCE. Then, we test how the mirror map $\psi$ effects the performance of MPO.

**Performance Comparison** We compare the convergence rate of MPO with REINFORCE and VPG empirically on the *Short Corridor with Switched Actions* domain (Chapter 13, Sutton & Barto (2018); We provide some details in Appendix B). The task is to estimate the value function of state $\mathbf{s}_1$, $V(\mathbf{s}_1) = G_0 \approx -11.6$.

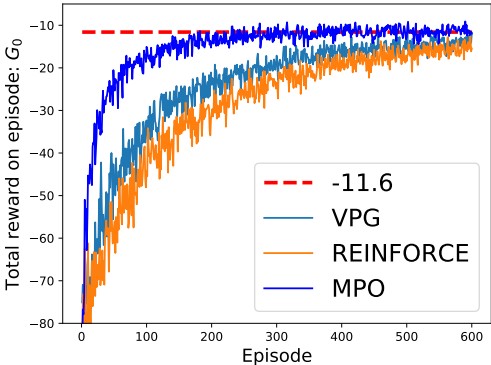

Figure 1: Comparison the performance of MPO with REINFORCE and VPG on the short-corridor grid world domain.

In this experiment, we use features $\phi(s, \texttt{right}) = [1, 0]^\top$ and $\phi(s, \texttt{left}) = [0, 1]^\top$, $s \in \mathcal{S}$. Let $L_\theta(s, a) = \phi^\top(s, a)\theta$, $(s, a) \in \mathcal{S} \times \mathcal{A}$, where $\mathcal{A} = \{\texttt{right}, \texttt{left}\}$. $\pi_\theta(a|s)$ is a exponential soft-max distribution defined as $\pi_\theta(a|s) = \dfrac{\exp\{L_\theta(s, a)\}}{\sum_{a' \in \mathcal{A}} \exp\{L_\theta(s, a')\}}$. The initial parameter $\theta_0 = \mathcal{U}[-0.5, 0.5]$, where $\mathcal{U}$ is uniform distribution.

Before we report the experimental results, it is necessary to explain why we only use VPG and REINFORCE as the baseline to compare with our MPO. VPG/REINFORCE is one of the most basic policy gradient methods in RL, and extensive modern policy-based algorithms are derived from VPG/REINFORCE. Our MPO is a novel framework via mirror map to learn the parameter, see (23). Thus, it is natural to compare with VPG and REINFORCE. The result in Figure 1 shows that MPO converges faster significantly and achieves a better performance than both REINFORCE and MPO.

**Effect of Mirror Map $\psi$** We use $\psi = \ell_p$-norm to test how the mirror map affects the performance of MPO. Particularly, the iteration (23) reduces to gradient descent update if $\psi = \ell_2$-norm. For $\psi = \ell_p$-norm, Eq.(23) has a closed implementation. Let $\psi^*(y) = (\sum_{i=1}^{n} |y_i|^q)^{\frac{1}{q}}$ be the conjugate map of $\psi$,

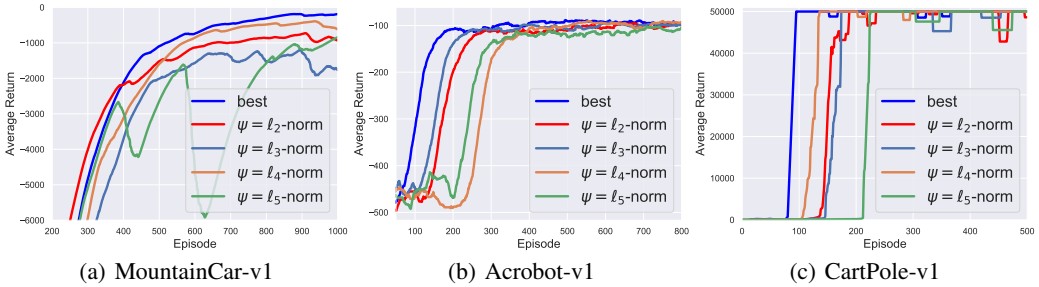

(a) MountainCar-v1         (b) Acrobot-v1         (c) CartPole-v1

Figure 2: Comparison of the empirical performance of `MPO` between non-Euclidean distance ($p \neq 2$) and Euclidean distance ($p = 2$) on standard domains: MountainCar, Acrobot and CartPole.

where $p^{-1} + q^{-1} = 1$, $p, q > 1$. According to (Beck & Teboulle, 2003), (23) is equivalent to

$$\theta_{k+1} = \nabla \psi^*(\nabla \psi(\theta_k) + \alpha_k \hat{g}_k),$$

where $\nabla \psi_j(x)$ and $\nabla \psi_j^*(y)$ are *p-norm link functions* (Gentile, 2003): $\nabla \psi_j(x) = \frac{\text{sign}(x_j)|x_j|^{p-1}}{\|x\|_p^{p-2}}, \nabla \psi_j^*(y) = \frac{\text{sign}(y_j)|y_j|^{q-1}}{\|y\|_q^{q-2}}$, and $j$ is coordinate index of the vector $\nabla \psi$, $\nabla \psi^*$.

To compare fairly, we use the same random seed, and let $p$ run in $[P] = \{1.1, 1.2, \cdots, 1.9, \mathbf{2}, 3, 4, 5\}$. For the non-Euclidean distance case, we only show $p = 3, 4, 5$, and "best", where $p$ = "best" value is that case it achieves the best performance among the set $[P]$. For the limitation of space, we provide more details of experiments in Appendix D.1.

The result in Figure 2 shows that the best method is produced by non-Euclidean distance, not the Euclidean distance. The traditional policy gradient methods such as `REINFORCE`, `VPG`, and `DPG` are all the algorithms update parameters in Euclidean distance. This simple experiment gives us some lights that one can create better algorithms by combining the existing approaches with non-Euclidean distance, which is an interesting direction, and we left it as future work.

## 6.2 EVALUATE VRMPO ON CONTINUOUS CONTROL TASKS

In this section, we compare `VRMPO` on the MuJoCo continuous control tasks (Todorov et al., 2012) from OpenAI Gym (Brockman et al., 2016). We compare `VRMPO` with `DDPG` (Lillicrap et al., 2016), `PPO` (Schulman et al., 2017), `TRPO` (Schulman et al., 2015), and `TD3` (Fujimoto et al., 2018). For fairness, all the setups mentioned above share the same network architecture that computes the policy and state value. We run all the algorithms with ten random seeds. The results of max-average epoch return are present in Table 2, and return curves are shown in Figure 3. For the limitation of space, we present all the details of experiments and some practical tricks for the implementation of `VRMPO` in Appendix D.2-D.5; in this section, we only offer our experimental results. We evaluate the performance of `VRMPO` by the following three aspects: score performance, the stability of training, and variance.

**Score Performance Comparison** From the results of Figure 3 and Table 2, overall, `VRMPO` outperforms the baseline algorithms in both final performance and learning process. Our `VRMPO` also learns considerably faster with better performance than the popular `TD3` on Walker2d-v2, HalfCheetah-v2, Hopper-v2, InvDoublePendulum-v2, and Reacher-v2 domains. On the InvDoublePendulum-v2 task, our `VRMPO` has only a small advantage over other algorithms. This is because the InvPendulum-v2 task is relatively easy. The advantage of our `VRMPO` becomes more powerful when the task is more difficult. It is worth noticing that on the HalfCheetah-v2 domain, our `VRMPO` achieves a significant max-average score 16000+, which outperforms far more than the second-best score 11781.

**Stability** The stability of an algorithm is also an important topic in RL. Although `DDPG` exploits the off-policy sample, which promotes its efficiency in stable environments, `DDPG` is unstable on the Reacher-v2 task, while our `VRMPO` learning faster significantly with lower variance. `DDPG` fails to make any progress on InvDoublePendulum-v2 domain, and the result is corroborated by the work (Dai et al., 2018). Although `TD3` takes the minimum value between a pair of critics to limit overes-

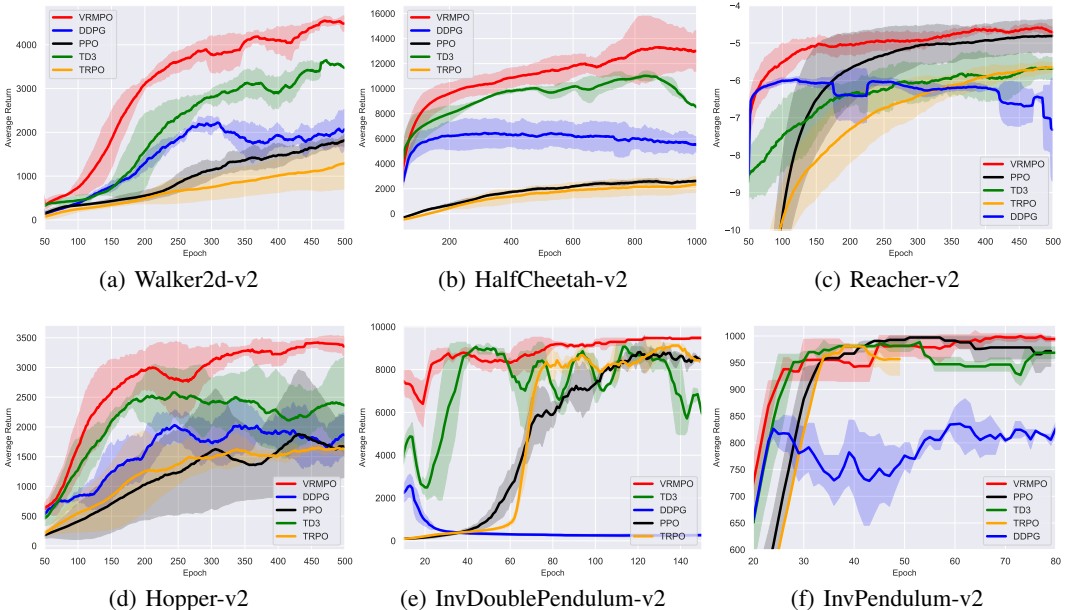

Figure 3: Learning curves for continuous control tasks. The shaded region represents the standard deviation of the score over the best three trials. Curves are smoothed uniformly for visual clarity.

| Environment | Our VRMPO | TD3 | DDPG | PPO | TRPO |
|---|---|---|---|---|---|
| Walker2d-v2 | 5251.83 | 4887.85 | **5795.13** | 3905.99 | 3636.59 |
| HalfCheetah-v2 | **16095.51** | 11781.07 | 8616.29 | 3542.60 | 3325.23 |
| Reacher-v2 | **-0.49** | -1.47 | -1.55 | **-0.44** | -0.66 |
| Hopper-v2 | **3751.43** | 3482.06 | 3558.69 | 3609.65 | 3578.06 |
| InvDoublePendulum-v2 | **9359.82** | 9248.27 | 6958.42 | 9045.86 | 9151.56 |
| InvPendulum-v2 | **1000.00** | **1000.00** | 907.81 | **1000.00** | **1000.00** |

Table 2: Max-average return over 500 epochs, where we run 5000 iterations for each epoch. Maximum value for each task is bolded.

timation, it learns severely fluctuating in the InvertedDoublePendulum-v2 environment. In contrast, our VRMPO is consistently reliable and effective in different tasks.

**Variance Comparison** As we can see from the results in Figure 3, our VRMPO converges with a considerably low variance in the Hopper-v2, InvDoublePendulum-v2, and Reacher-v2. Although the asymptotic variance of VRMPO is slightly larger than other algorithms in HalfCheetah-v2, the final performance of VRMPO outperforms all the baselines significantly. The result in Figure 3 also implies conducting a proper gradient estimator not only reduce the variance of the score during the learning but speed the convergence of training.

## 7 CONCLUSION

In this paper, we propose the mirror policy optimization (MPO) by estimating the policy gradient via dynamic batch-size of historical gradient information. Results show that making use of historical gradients to estimate policy gradient is more effective to speed convergence. We also propose a variance reduction implementation for MPO: VRMPO, and prove the complexity of VRMPO achieves $\mathcal{O}(\epsilon^{-3})$. To our best knowledge, VRMPO matches the best-known sample complexity so far. Finally, we evaluate the performance of VRMPO on the MuJoCo continuous control tasks, results show that VRMPO outperforms or matches several state-of-art algorithms DDPG, TRPO, PPO, and TD3.

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

## A  PROOF OF THEOREM 2

Let $f(x)$ be a $L$-smooth function defined on $\mathbb{R}^n$, i.e $\|\nabla f(x) - \nabla f(y)\| \leq L\|x - y\|$. Then, for $\forall x, y \in \mathbb{R}^n$, the following holds

$$\|f(x) - f(y) - \langle \nabla f(y), x - y \rangle\| \leq \frac{L}{2}\|x - y\|^2. \tag{33}$$

The following Lemma 2 is useful for our proof.

**Lemma 2** (Ghadimi et al. (2016), Lemma 1 and Proposition 1). *Let $\mathcal{X}$ be a closed convex set in $\mathbb{R}^d$, $h : \mathcal{X} \to \mathbb{R}$ be a convex function, but possibly nonsmooth, and $D_\psi : \mathcal{X} \times \mathcal{X} \to \mathbb{R}$ is Bregman divergence. Moreover, define*

$$x^+ = \arg\min_{u \in \mathcal{X}} \left\{ \langle g, u \rangle + \frac{1}{\eta}D_\psi(u, x) + h(u) \right\}$$
$$P_\mathcal{X}(x, g, \eta) = \frac{1}{\eta}(x - x^+), \tag{34}$$

*where $g \in \mathbb{R}^d$, $x \in \mathcal{X}$, and $\eta > 0$. Then, the following statement holds*

$$\langle g, P_\mathcal{X}(x, g, \eta) \rangle \geq \zeta \|P_\mathcal{X}(x, g, \eta)\|^2 + \frac{1}{\eta}[h(x^+) - h(x)], \tag{35}$$

*where $\zeta$ is a positive constant determined by $\psi$ (i.e. $\psi$ is a a continuously-differentiable and $\zeta$-strictly convex function) that satisfying $\langle x - y, \nabla\psi(x) - \nabla\psi(y) \rangle \geq \zeta \|x - y\|^2$. Moreover, for any $g_1, g_2 \in \mathbb{R}^d$, the following statement holds*

$$\|P_\mathcal{X}(x, g_1, \eta) - P_\mathcal{X}(x, g_2, \eta)\| \leq \frac{1}{\zeta}\|g_1 - g_2\|. \tag{36}$$

**Theorem 2** (Convergence Rate of Algorithm 1) *Under Assumption 1, and the total trajectories are $\{\tau_k\}_{k=1}^N$. Consider the sequence $\{\theta_k\}_{k=1}^N$ generated by Algorithm 1, and the output $\tilde{\theta}_N = \theta_n$ follows the distribution of Eq.(16). Let $0 < \alpha_k < \frac{\zeta}{L}$, $\ell(g, u) = \langle g, u \rangle$, the term $L$ and $\sigma$ are defined in Eq.(5) and Eq.(6) correspondingly. Let $\hat{g}_k = \frac{1}{k}\sum_{i=1}^k g_i$, where $g_i = \sum_{t=0}^{H_{\tau_i}} \nabla_\theta \log \pi_\theta(a_t|s_t)R(\tau_i)|_{\theta=\theta_i}$. Then we have*

$$\mathbb{E}[\|\mathcal{G}_{\alpha_n,\ell(-g_n,\theta_n)}^\psi(\theta_n)\|^2] \leq \frac{(J(\theta^*) - J(\theta_1)) + \frac{\sigma^2}{\zeta}\sum_{k=1}^N \frac{\alpha_k}{k}}{\sum_{k=1}^N (\zeta\alpha_k - L\alpha_k^2)}.$$

*Proof.* **(Proof of Theorem 2)**

Let $\mathcal{T} = \{\tau_k\}_{k=1}^N$ be the trjecories generated by the differentiable parametric policy $\pi_\theta$. At each terminal end of a trajectory $\tau_k = \{s_t, a_t, r_{t+1}\}_{t=0}^{H_{\tau_k}} \in \mathcal{T}$, let

$$g_k = \sum_{t=0}^{H_{\tau_k}} \nabla_\theta \log \pi_\theta(a_t|s_t)R(\tau_k)|_{\theta=\theta_k}, \quad \hat{g}_k = \frac{1}{k}\sum_{i=1}^k g_i,$$

according to Algorithm 1, at the terminal end of $k$-th episode, $k = 1, 2, \cdots, N$, the following holds,

$$\theta_{k+1} = \arg\min_\theta \left\{ \langle -\hat{g}_k, \theta \rangle + \frac{1}{\alpha_k}D_\psi(\theta, \theta_k) \right\} = \mathcal{M}_{\alpha_k,\ell(-\hat{g}_k,\theta)}^\psi(\theta_k).$$

To simplify expression, let $\mathcal{J}(\theta) = -J(\theta)$, then $\mathcal{J}(\theta)$ is $L$-smooth, from Eq.(33), we have

$$\mathcal{J}(\theta_{k+1}) \leq \mathcal{J}(\theta_k) + \left\langle \nabla_\theta \mathcal{J}(\theta)\big|_{\theta=\theta_k}, \theta_{k+1} - \theta_k \right\rangle + \frac{L}{2}\left\|\theta_{k+1} - \theta_k\right\|^2$$

$$= \mathcal{J}(\theta_k) - \alpha_k \left\langle \nabla\mathcal{J}(\theta_k), \mathcal{G}_{\alpha_k,\ell(-\hat{g}_k,\theta_k)}^\psi(\theta_k) \right\rangle + \frac{L\alpha_k^2}{2}\left\|\mathcal{G}_{\alpha_k,\ell(-\hat{g}_k,\theta_k)}^\psi(\theta_k)\right\|^2$$

$$= \mathcal{J}(\theta_k) - \alpha_k \left\langle \hat{g}_k, \mathcal{G}_{\alpha_k,\ell(-\hat{g}_k,\theta_k)}^\psi(\theta_k) \right\rangle + \frac{L\alpha_k^2}{2}\left\|\mathcal{G}_{\alpha_k,\ell(-\hat{g}_k,\theta_k)}^\psi(\theta_k)\right\|^2$$

$$+ \alpha_k \left\langle \epsilon_k, \mathcal{G}_{\alpha_k,\ell(-\hat{g}_k,\theta_k)}^\psi(\theta_k) \right\rangle,$$

where $\epsilon_k = -\hat{g}_k - (-\nabla J(\theta_k)) = -\hat{g}_k - \nabla \mathcal{J}(\theta_k)$. By Eq.(34) and let $h(x) \equiv 0$ and $\eta = \alpha$, then $P_{\mathcal{X}}(\theta, g, \alpha) = \mathcal{G}^{\psi}_{\alpha,\ell(g,\theta)}(\theta)$. Furthermore, by Eq.(35), let $\eta = \alpha_k$ and $g = -\hat{g}_k$, then we have

$$\mathcal{J}(\theta_{k+1}) \leq \mathcal{J}(\theta_k) - \alpha_k \zeta \left\|\mathcal{G}^{\psi}_{\alpha_k,\ell(-\hat{g}_k,\theta_k)}(\theta_k)\right\|^2 + \frac{L\alpha_k^2}{2}\left\|\mathcal{G}^{\psi}_{\alpha_k,\ell(-\hat{g}_k,\theta_k)}(\theta_k)\right\|^2$$
$$+ \alpha_k\left\langle\epsilon_k, \mathcal{G}^{\psi}_{\alpha_k,\ell(-\hat{g}_k,\theta_k)}(\theta_k)\right\rangle$$
$$= \mathcal{J}(\theta_k) - \alpha_k \zeta \left\|\mathcal{G}^{\psi}_{\alpha_k,\ell(-\hat{g}_k,\theta_k)}(\theta_k)\right\|^2 + \frac{L\alpha_k^2}{2}\left\|\mathcal{G}^{\psi}_{\alpha_k,\ell(-\hat{g}_k,\theta_k)}(\theta_k)\right\|^2$$
$$+ \alpha_k\left\langle\epsilon_k, \mathcal{G}^{\psi}_{\alpha_k,\ell(-\nabla J(\theta_k),\theta_k)}(\theta_k)\right\rangle$$
$$+ \alpha_k\left\langle\epsilon_k, \mathcal{G}^{\psi}_{\alpha_k,\ell(-\hat{g}_k,\theta_k)}(\theta_k) - \mathcal{G}^{\psi}_{\alpha_k,\ell(-\nabla J(\theta_k),\theta_k)}(\theta_k)\right\rangle. \tag{37}$$

Rearrange Eq.(37), we have

$$\mathcal{J}(\theta_{k+1}) \leq \mathcal{J}(\theta_k) - \left(\zeta\alpha_k - \frac{L\alpha_k^2}{2}\right)\left\|\mathcal{G}^{\psi}_{\alpha_k,\ell(-\hat{g}_k,\theta_k)}(\theta_k)\right\|^2 + \alpha_k\left\langle\epsilon_k, \mathcal{G}^{\psi}_{\alpha_k,\ell(-\nabla J(\theta_k),\theta_k)}(\theta_k)\right\rangle$$
$$+ \alpha_k\|\epsilon_k\|\left\|\mathcal{G}^{\psi}_{\alpha_k,\ell(-\hat{g}_k,\theta_k)}(\theta_k) - \mathcal{G}^{\psi}_{\alpha_k,\ell(-\nabla J(\theta_k),\theta_k)}(\theta_k)\right\|.$$

By Eq.(36), let $x = \theta_k, g_1 = -\hat{g}_k, g_2 = -\nabla J(\theta_k), h(x) \equiv 0$, then the following statement holds

$$\mathcal{J}(\theta_{k+1})$$
$$\leq \mathcal{J}(\theta_k) - \left(\zeta\alpha_k - \frac{L\alpha_k^2}{2}\right)\left\|\mathcal{G}^{\psi}_{\alpha_k,\ell(-\hat{g}_k,\theta_k)}(\theta_k)\right\|^2 + \alpha_k\left\langle\epsilon_k, \mathcal{G}^{\psi}_{\alpha_k,\ell(-\nabla J(\theta_k),\theta_k)}(\theta_k)\right\rangle + \frac{\alpha_k}{L}\|\epsilon_k\|^2. \tag{38}$$

Summing the above Eq.(38) from $k = 1$ to $N$ and with the condition $\alpha_k \leq \frac{\zeta}{L}$, we have the following statement

$$\sum_{k=1}^{N}\left(\zeta\alpha_k - L\alpha_k^2\right)\left\|\mathcal{G}^{\psi}_{\alpha_k,\ell(-\hat{g}_k,\theta_k)}(\theta_k)\right\|^2$$
$$\leq \sum_{k=1}^{N}\left(\zeta\alpha_k - \frac{L\alpha_k^2}{2}\right)\left\|\mathcal{G}^{\psi}_{\alpha_k,\ell(-\hat{g}_k,\theta_k)}(\theta_k)\right\|^2$$
$$\leq \sum_{k=1}^{N}\left[\alpha_k\left\langle\epsilon_k, \mathcal{G}^{\psi}_{\alpha_k,\ell(-\nabla J(\theta_k),\theta_k)}(\theta_k)\right\rangle + \frac{\alpha_k}{\zeta}\|\epsilon_k\|^2\right] + \mathcal{J}(\theta_1) - \mathcal{J}(\theta_{k+1})$$
$$\leq \sum_{k=1}^{N}\left[\alpha_k\left\langle\epsilon_k, \mathcal{G}^{\psi}_{\alpha_k,\ell(-\nabla J(\theta_k),\theta_k)}(\theta_k)\right\rangle + \frac{\alpha_k}{\zeta}\|\epsilon_k\|^2\right] + \mathcal{J}(\theta_1) - \mathcal{J}^*. \tag{39}$$

Recall

$$g_k = \sum_{t=0}^{H_{\tau_k}}\nabla_\theta\log\pi_\theta(a_t|s_t)R(\tau_k), \hat{g}_k = \frac{1}{k}\sum_{i=1}^{k}g_i,$$

by policy gradient theorem

$$\mathbb{E}[-g_k] = \mathbb{E}[-\hat{g}_k] = -\nabla J(\theta_k) = \nabla\mathcal{J}(\theta_k). \tag{40}$$

Let $\mathcal{F}_k$ be the $\sigma$-field generated by all random variables defined before round $k$, $\tilde{\epsilon}_k = g_k - \nabla\mathcal{J}(\theta_k)$ then the Eq.(40) implies: for $k = 1, \cdots, N$,

$$\mathbb{E}\left[\left\langle\epsilon_k, \mathcal{G}^{\psi}_{\alpha_k,\ell(-\nabla J(\theta_k),\theta_k)}(\theta_k)\right\rangle\Big|\mathcal{F}_{k-1}\right] = \mathbb{E}\left[\left\langle\tilde{\epsilon}_k, \mathcal{G}^{\psi}_{\alpha_k,\ell(-\nabla J(\theta_k),\theta_k)}(\theta_k)\right\rangle\Big|\mathcal{F}_{k-1}\right] = 0.$$

Let $\delta_s = \sum_{t=1}^{s}\tilde{\epsilon}_t$, noticing that for $s = 1, \cdots, k$,

$$\mathbb{E}[\langle\delta_s, \tilde{\epsilon}_{s+1}\rangle|\delta_s] = 0. \tag{41}$$

Furthermore, the following statement holds

$$\mathbb{E}[\|\delta_k\|^2] = \mathbb{E}\left[\|\delta_{k-1}\|^2 + 2\left\langle\delta_{k-1}, \tilde{\epsilon}_k\right\rangle + \|\tilde{\epsilon}_t\|^2\right] \overset{(41)}{=} \mathbb{E}\left[\|\delta_{k-1}\|^2 + \|\tilde{\epsilon}_t\|^2\right] = \cdots = \sum_{t=1}^{k}\mathbb{E}\|\tilde{\epsilon}_t\|^2.$$

$$(42)$$

By Lemma 1 and Eq.(42), we have

$$\mathbb{E}[\|\epsilon_k\|^2] = \frac{1}{k^2}\sum_{t=1}^{k}\mathbb{E}\|\tilde{\epsilon}_t\|^2 \leq \frac{\sigma^2}{k}. \qquad (43)$$

Taking Eq.(43) in to Eq.(39), and taking expections w.r.t $\mathcal{F}_N$, we have

$$\sum_{k=1}^{N}\left(\zeta\alpha_k - L\alpha_k^2\right)\mathbb{E}\left[\left\|\mathcal{G}_{\alpha_k,\ell(-\hat{g}_k,\theta_k)}^{\psi}(\theta_k)\right\|^2\right] \leq \mathcal{J}\theta_1) - \mathcal{J}^* + \frac{\sigma^2}{\zeta}\sum_{k=1}^{N}\frac{\alpha_k}{k}.$$

Now, consider the output $\tilde{\theta}_N = \theta_n$ follows the distribution of Eq.(16), we have

$$\mathbb{E}[\|\mathcal{G}_{\alpha_n,\ell(-g_n,\theta_n)}^{\psi}(\theta_n)\|^2] \leq \frac{\left(J(\theta^*) - J(\theta_1)\right) + \frac{\sigma^2}{\zeta}\sum_{k=1}^{N}\frac{\alpha_k}{k}}{\sum_{k=1}^{N}(\zeta\alpha_k - L\alpha_k^2)}.$$

Particularly, if the step-size $\alpha_k$ is fixed to a constant: $\zeta/2L$, then

$$\mathbb{E}[\|\mathcal{G}_{\alpha_n,\ell(-\hat{g}_n,\theta_n)}^{\psi}(\theta_n)\|^2] \leq \frac{4L\left(J(\theta^*) - J(\theta_1)\right) + 2\sigma^2\sum_{k=1}^{N}\frac{1}{k}}{N\zeta^2}.$$

Recall the following estimation

$$\sum_{k=1}^{N}\frac{1}{k} = \ln N + C + o(1),$$

where $C$ is the Euler constant—a positive real number and $o(1)$ is infinitesimal. Thus the overall convergence rate reaches $\mathcal{O}(\frac{\ln N}{N})$ as

$$\mathbb{E}[\|\mathcal{G}_{\alpha_n,\ell(-\hat{g}_n,\theta_n)}^{\psi}(\theta_n)\|^2] \leq \frac{4LD_J^2 + 2\sigma\sum_{k=1}^{N}\frac{1}{k}}{N\zeta} = \mathcal{O}(\frac{\ln N}{N}).$$

$$\square$$

## B    SHORT CORRIDOR WITH SWITCHED ACTIONS

Consider the small corridor grid world which contains three sates $\mathcal{S} = \{1, 2, 3\}$. The reward is $-1$ per step. In each of the three nonterminal states there are only two actions, `right` and `left`. These actions have their usual consequences in the state $1$ and state $3$ (`left` causes no movement in the first state), but in the state $2$ they are reversed. so that `right` moves to the left and `left` moves to the right.

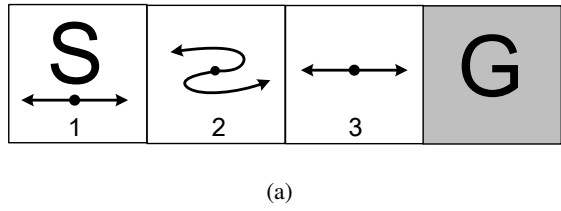

(a)

Figure 4:  Short corridor with switched actions (Chapter 13, (Sutton & Barto, 2018)).

An action-value method with $\epsilon$-greedy action selection is forced to choose between just two policies: choosing `right` with high probability $1 - \frac{\epsilon}{2}$ on all steps or choosing `left` with the same high probability on all time steps. If $\epsilon = 0.1$, then these two policies achieve a value (at the start state) of less than $-44$ and $-82$, respectively, as shown in the following graph.

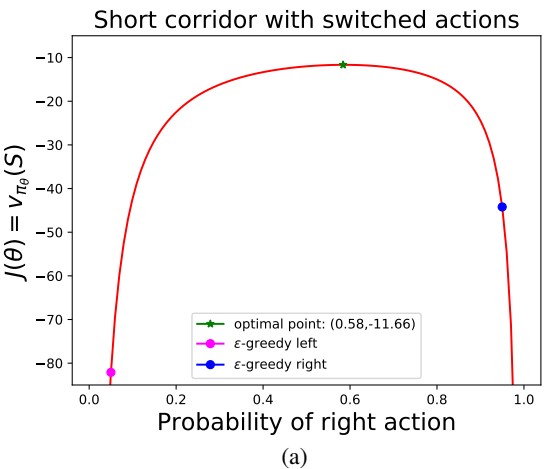

(a)

Figure 5:  Short corridor with switched actions (Chapter 13, (Sutton & Barto, 2018)).

A method can do significantly better if it can learn a specific probability with which to select right. The best probability is about $0.58$, which achieves a value of about $-11.6$.

## C  PROOF OF THEOREM 3

We need the following lemmas to prove the convergence result.

**Lemma 3** (Lemma1 (Fang et al., 2018)). *Under Assumption 1, $G_{k,t}$ is generated according to (25), $\theta_{k,t}$ is generated according to (26), then for any $0 \leq t \leq m$, the following holds*

$$\mathbb{E}[\|G_{k,t} - \nabla \mathcal{J}(\theta_{k,t})\|^2] \leq \frac{L^2}{N_2}\mathbb{E}[\|\theta_{k,t} - \theta_{k,t-1}\|^2] + \mathbb{E}[\|G_{k,t-1} - \mathcal{J}(\theta_{k,t-1})\|^2]. \quad (44)$$

*Telescoping Eq.(3) over $t$ from 1 to the time $t$, then the following holds*

$$\mathbb{E}[\|G_{k,t} - \nabla \mathcal{J}(\theta_{k,t})\|^2] \leq \sum_{i=1}^{t} \frac{L^2}{N_2}\mathbb{E}[\|\theta_{k,i+1} - \theta_{k,i}\|^2] + \mathbb{E}[\|G_{k-1,0} - \nabla \mathcal{J}(\tilde{\theta}_{k-1})\|^2]. \quad (45)$$

**Lemma 4.** *Let $\zeta > \frac{5}{32}$, the batch size of the trajectories of outer loop $N_1 = \dfrac{\left(\frac{1}{8L\zeta^2} + \frac{1}{2(\zeta - \frac{5}{32})}\left(1 + \frac{1}{32\zeta^2}\right)\right)\sigma^2}{\epsilon^2}$, the iteration times of inner loop $m - 1 = N_2 = \dfrac{\sqrt{\left(\frac{1}{8L\zeta^2} + \frac{1}{2(\zeta - \frac{5}{32})}\left(1 + \frac{1}{32\zeta^2}\right)\right)}\sigma}{\epsilon}$, and step size $\alpha_k = \frac{1}{4L}$. For each $k$ and $t$, $G_{k,0}$ and $\theta_{k,0}$ are generated by Algorithm 2, then the following holds,*

$$\mathbb{E}\|\nabla \mathcal{J}(\theta_{k,0}) - G_{k,0}\|^2 \leq \left(\frac{1}{8L\zeta^2} + \frac{1}{2(\zeta - \frac{5}{32})}\left(1 + \frac{1}{32\zeta^2}\right)\right)^{-1}\epsilon^2. \quad (46)$$

*Proof.*

$$\mathbb{E}\|\nabla \mathcal{J}(\theta_{k,0}) - G_{k,0}\|^2 = \mathbb{E}\|\nabla J(\theta_{k,0}) - \frac{1}{N_1}\sum_{i=1}^{N_1} g(\tau_i|\theta_{k,0})\|^2 \quad (47)$$

$$= \frac{1}{N_1^2}\sum_{i=1}^{N_1}\mathbb{E}\|\nabla J(\theta_{k,0}) - g(\tau_i|\theta_{k,0})\|^2 \quad (48)$$

$$\overset{(6)}{\leq} \frac{\sigma^2}{N_1} \quad (49)$$

$$= \underbrace{\left(\frac{1}{8L\zeta^2} + \frac{1}{2(\zeta - \frac{5}{32})}\left(1 + \frac{1}{32\zeta^2}\right)\right)^{-1}\epsilon^2}_{\overset{\text{def}}{=}\epsilon_1^2}. \quad (50)$$

$\square$

**Theorem 3** (Convergence Rate of VRMPO) *The sequence $\{\tilde{\theta}_k\}_{k=1}^{K}$ is generated according to Algorithm 2. Under Assumption 1, let $\zeta > \frac{5}{32}$, the batch size of the trajectories of outer loop $N_1 = \left(\frac{1}{8L\zeta^2} + \frac{1}{2(\zeta - \frac{5}{32})}\left(1 + \frac{1}{32\zeta^2}\right)\right)\frac{\sigma^2}{\epsilon^2}$, the iteration times of inner loop $m - 1 = N_2 = \sqrt{\left(\frac{1}{8L\zeta^2} + \frac{1}{2(\zeta - \frac{5}{32})}\left(1 + \frac{1}{32\zeta^2}\right)\right)}\frac{\sigma}{\epsilon}$, the iteration times of outer loop $K = \frac{8L}{(m-1)(\zeta - \frac{5}{32})}\left(1 + \frac{1}{16\zeta^2}\right)\frac{1}{\epsilon^2}$, and step size $\alpha_k = \frac{1}{4L}$. Then, Algorithm 2 outputs the point $\tilde{\theta}_K$ achieves*

$$\mathbb{E}[\|\mathcal{G}_{\alpha, \langle -\nabla J(\tilde{\theta}_K), \theta \rangle}^{\psi}(\tilde{\theta}_K)\|] \leq \epsilon. \quad (51)$$

*Proof.* (**Proof of Theorem 3**)

By the definition of Bregman grdient mapping in Eq.(10) and iteration (26), let $\alpha_k = \alpha$, we have

$$\frac{1}{\alpha}(\theta_{k,t} - \theta_{k,t+1}) \overset{(26)}{=} \underbrace{\frac{1}{\alpha}\left(\theta_{k,t} - \arg\min_u\{\langle G_{k,t}, u\rangle + \frac{1}{\alpha_k}D_\psi(u, \theta_{k,t})\}\right)}_{g_{k,t}} \overset{(10)}{=} \mathcal{G}^\psi_{\alpha, \langle G_{k,t}, u\rangle}(\theta_{k,t}),$$

(52)

where we introduce $g_{k,t}$ to simplify notations.

**Step 1: Analyze the inner loop of Algorithm 2** Now, we analyze the inner loop of Algorithm 2. In this step, our goal is to prove

$$\mathbb{E}[\mathcal{J}(\tilde{\theta}_k)] - \mathbb{E}[\mathcal{J}(\tilde{\theta}_{k-1})] \le -\sum_{t=1}^{m-1}\left(\eta\mathbb{E}[\|g_{k,t}\|^2] - \frac{\alpha}{2}\epsilon^2\right).$$

In fact,

$$
\begin{aligned}
\mathcal{J}(\theta_{k,t+1}) &\overset{(33)}{\le} \mathcal{J}(\theta_{k,t}) + \langle\nabla\mathcal{J}(\theta_{k,t}), \theta_{k,t+1} - \theta_{k,t}\rangle + \frac{L}{2}\|\theta_{k,t+1} - \theta_{k,t}\|^2 \\
&\overset{(52)}{=} \mathcal{J}(\theta_{k,t}) - \alpha\langle\nabla\mathcal{J}(\theta_{k,t}), g_{k,t}\rangle + \frac{L\alpha^2}{2}\|g_{k,t}\|^2 \\
&= \mathcal{J}(\theta_{k,t}) - \alpha\langle\nabla\mathcal{J}(\theta_{k,t}) - G_{k,t}, g_{k,t}\rangle - \alpha\langle G_{k,t}, g_{k,t}\rangle + \frac{L\alpha^2}{2}\|g_{k,t}\|^2 \\
&\le \mathcal{J}(\theta_{k,t}) + \frac{\alpha}{2}\|\nabla\mathcal{J}(\theta_{k,t}) - G_{k,t}\|^2 - \alpha\langle G_{k,t}, g_{k,t}\rangle + \left(\frac{L\alpha^2}{2} + \frac{\alpha}{2}\right)\|g_{k,t}\|^2 \quad (53) \\
&\overset{(35)}{\le} \mathcal{J}(\theta_{k,t}) + \frac{\alpha}{2}\|\nabla\mathcal{J}(\theta_{k,t}) - G_{k,t}\|^2 - \zeta\alpha\|g_{k,t}\|^2 + \left(\frac{L\alpha^2}{2} + \frac{\alpha}{2}\right)\|g_{k,t}\|^2, \quad (54)
\end{aligned}
$$

Eq.(53) holds due to the Cauchy-Schwarz inequality $|\langle\mathbf{u}, \mathbf{v}\rangle| \le \|\mathbf{u}\|\|\mathbf{v}\| \le \frac{1}{2}(\|\mathbf{u}\|^2 + \|\mathbf{v}\|^2)$ for any $\mathbf{u}, \mathbf{v} \in \mathbb{R}^n$. Eq.(54) holds if $h \equiv 0$ by Eq.(35).

Taking expectation on both sides of Eq(54), we have

$$
\begin{aligned}
\mathbb{E}[\mathcal{J}(\theta_{k,t+1})] &\le \mathbb{E}[\mathcal{J}(\theta_{k,t})] + \frac{\alpha}{2}\mathbb{E}\left[\|\nabla\mathcal{J}(\theta_{k,t}) - G_{k,t}\|^2\right] - \left(\zeta\alpha - \frac{L\alpha^2}{2} - \frac{\alpha}{2}\right)\mathbb{E}\left[\|g_{k,t}\|^2\right] \\
&\le \mathbb{E}[\mathcal{J}(\theta_{k,t})] + \frac{\alpha}{2}\sum_{i=1}^t \frac{L^2}{N_2}\mathbb{E}\|\theta_{k,i+1} - \theta_{k,i}\|^2 \\
&\quad + \frac{\alpha}{2}\mathbb{E}\|G_{k-1,0} - \nabla\mathcal{J}(\theta_{k-1,0})\|^2 - \left(\zeta\alpha - \frac{L\alpha^2}{2} - \frac{\alpha}{2}\right)\mathbb{E}\|g_{k,t}\|^2 \quad (55)
\end{aligned}
$$

Eq.(55) holds due to Lemma 3.

By Lemma 4, Eq.(55) and Eq.(52), we have

$$\mathbb{E}[\mathcal{J}(\theta_{k,t+1})] \le \mathbb{E}[\mathcal{J}(\theta_{k,t})] + \frac{\alpha^3 L^2}{2N_2}\sum_{i=1}^t \mathbb{E}\left[\|g_{k,i}\|^2\right] + \frac{\alpha\epsilon_1^2}{2} - \left(\zeta\alpha - \frac{L\alpha^2}{2} - \frac{\alpha}{2}\right)\mathbb{E}\|g_{k,t}\|^2.$$

Recall the parameter $\tilde{\theta}_{k-1} = \theta_{k-1,m}$ is generated by the last time of $(k-1)$-th episode, we now consider the following equation

$$\mathbb{E}[\mathcal{J}(\theta_{k,t+1})] - \mathbb{E}[\mathcal{J}(\tilde{\theta}_{k-1})]$$

$$\leq \frac{\alpha^3 L^2}{2N_2} \sum_{j=1}^{t} \sum_{i=1}^{j} \mathbb{E}\|g_{k,i}\|^2 + \frac{\alpha}{2} \sum_{j=1}^{t} \epsilon_1^2 - \left(\zeta\alpha - \frac{L\alpha^2}{2} - \frac{\alpha}{2}\right) \sum_{j=1}^{t} \mathbb{E}\|g_{k,j}\|^2$$

$$\leq \frac{\alpha^3 L^2}{2N_2} \sum_{j=1}^{t} \sum_{i=1}^{t} \mathbb{E}\|g_{k,i}\|^2 + \frac{\alpha}{2} \sum_{j=1}^{t} \epsilon_1^2 - \left(\zeta\alpha - \frac{L\alpha^2}{2} - \frac{\alpha}{2}\right) \sum_{j=1}^{t} \mathbb{E}\|g_{k,j}\|^2$$

$$= \frac{\alpha^3 L^2 t}{2N_2} \sum_{i=1}^{t} \mathbb{E}\|g_{k,i}\|^2 + \frac{\alpha}{2} \sum_{j=1}^{t} \epsilon_1^2 - \left(\zeta\alpha - \frac{L\alpha^2}{2} - \frac{\alpha}{2}\right) \sum_{j=1}^{t} \mathbb{E}\|g_{k,j}\|^2$$

$$\leq \frac{\alpha^3 L^2 (m-1)}{2N_2} \sum_{i=1}^{t} \mathbb{E}\|g_{k,i}\|^2 + \frac{\alpha}{2} \sum_{j=1}^{t} \epsilon_1^2 - \left(\zeta\alpha - \frac{L\alpha^2}{2} - \frac{\alpha}{2}\right) \sum_{j=1}^{t} \mathbb{E}\|g_{k,j}\|^2 \qquad (56)$$

$$= \frac{\alpha}{2} \sum_{j=1}^{t} \epsilon_1^2 - \underbrace{\left(\zeta\alpha - \frac{L\alpha^2}{2} - \frac{\alpha}{2} - \frac{\alpha^3 L^2 (m-1)}{2N_2}\right)}_{\stackrel{\text{def}}{=}\eta=\frac{\zeta - \frac{5}{32}}{4L}} \sum_{j=1}^{t} \mathbb{E}\|g_{k,j}\|^2$$

$$= -\sum_{i=1}^{t} \left( \eta\mathbb{E}[\|g_{k,i}\|^2] - \frac{\alpha}{2} \underbrace{\left( \left( \frac{1}{8L\zeta^2} + \frac{1}{2(\zeta - \frac{5}{32})} \left(1 + \frac{1}{32\zeta^2}\right) \right) \right)^{-1} \epsilon^2}_{=\epsilon_1^2} \right), \qquad (57)$$

Eq.(56) holds due to $t \leq m-1$.

If $t = m-1$, then by the last Eq.(57) implies

$$\mathbb{E}[\mathcal{J}(\tilde{\theta}_k)] - \mathbb{E}[\mathcal{J}(\tilde{\theta}_{k-1})] \leq -\sum_{t=1}^{m-1} \left( \eta\mathbb{E}[\|g_{k,t}\|^2] - \frac{\alpha}{2}\epsilon_1^2 \right). \qquad (58)$$

### Step 2: Analyze the outer loop of Algorithm 2

We now consider the output of Algorithm 2,

$$\mathbb{E}[\mathcal{J}(\tilde{\theta}_K)] - \mathbb{E}[\mathcal{J}(\tilde{\theta}_0)] = \left( \mathbb{E}[\mathcal{J}(\tilde{\theta}_1)] - \mathbb{E}[\mathcal{J}(\tilde{\theta}_0)] \right) + \left( \mathbb{E}[\mathcal{J}(\tilde{\theta}_2)] - \mathbb{E}[\mathcal{J}(\tilde{\theta}_1)] \right)$$

$$+ \cdots + \left( \mathbb{E}[\mathcal{J}(\tilde{\theta}_K)] - \mathbb{E}[\mathcal{J}(\tilde{\theta}_{K-1})] \right)$$

$$\stackrel{(58)}{\leq} -\sum_{t=0}^{m-1} \left( \eta\mathbb{E}\|g_{1,t}\|^2 - \frac{\alpha}{2}\epsilon_1^2 \right) - \sum_{t=0}^{m-1} \left( \eta\mathbb{E}\|g_{2,t}\|^2 - \frac{\alpha}{2}\epsilon_1^2 \right)$$

$$- \cdots - \sum_{t=0}^{m-1} \left( \eta\mathbb{E}\|g_{K,t}\|^2 - \frac{\alpha}{2}\epsilon_1^2 \right)$$

$$= -\sum_{k=1}^{K} \sum_{t=0}^{m-1} \left( \eta\mathbb{E}\|g_{k,t}\|^2 - \frac{\alpha}{2}\epsilon_1^2 \right)$$

$$= -\sum_{k=1}^{K} \sum_{t=1}^{m-1} \left( \eta\mathbb{E}\|g_{k,t}\|^2 \right) + \frac{K\alpha}{2}\epsilon_1^2,$$

then we have

$$\sum_{k=1}^{K} \sum_{t=1}^{m-1} \left( \eta\mathbb{E}\|g_{k,t}\|^2 \right) \leq \mathbb{E}[\mathcal{J}(\tilde{\theta}_0)] - \mathcal{J}(\theta^*) + \frac{K(m-1)\alpha}{2}\epsilon_1^2. \qquad (59)$$

Recall the notation in Eq.(52)

$$g_{k,t} = \frac{1}{\alpha}(\theta_{k,t} - \arg\min_u\{\langle G_{k,t}, u\rangle + \frac{1}{\alpha}D_\psi(u, \theta_{k,t})\}) = \mathcal{G}^\psi_{\alpha,\langle G_{k,t},u\rangle}(\theta_{k,t}),$$

and we introduce following $\tilde{g}(\theta_{k,t})$ to simplify notations,

$$\tilde{g}(\theta_{k,t}) = \mathcal{G}^\psi_{\alpha,\langle -\nabla J(\theta_{k,t}),u\rangle}(\theta_{k,t}) \overset{\text{def}}{=} \tilde{g}_{k,t}$$
$$= \frac{1}{\alpha}\left(\theta_{k,t} - \arg\min_u\{\langle -\nabla J(\theta_{k,t}), u\rangle + \frac{1}{\alpha}D_\psi(u, \theta_{k,t})\}\right). \tag{60}$$

Then, the following holds

$$\mathbb{E}\|\tilde{g}_{k,t}\|^2 \leq \mathbb{E}\|g_{k,t}\|^2 + \mathbb{E}\|\tilde{g}_{k,t} - g_{k,t}\|^2$$
$$\overset{(36)}{\leq} \mathbb{E}\|g_{k,t}\|^2 + \frac{1}{\zeta^2}\mathbb{E}\|\nabla J(\theta_{k,t}) - G_{k,t}\|^2, \tag{61}$$

Eq.(61) holds due to the Eq.(36).

Let $\nu$ be the number that is selected randomly from $\{1, \cdots, (m-1)K\}$ which is the output of Algorihtm 2,for the convenience of proof the there is no harm in hypothesis that $\nu = k \cdot (m-1) + t$ and we denote the output $\theta_\nu = \theta_{k,t}$.

Now, we analyze above Eq.(61) and show it is bounded as following two parts (63) and (66)

$$\mathbb{E}\|g(\theta_\nu)\|^2 = \frac{1}{(m-1)K}\sum_{k=1}^K\sum_{t=1}^{m-1}\mathbb{E}\|g_{k,t}\|^2 \overset{(59)}{\leq} \frac{\mathbb{E}[J(\tilde{\theta}_0)] - J(\theta^*)}{(m-1)K\eta} + \frac{\alpha}{2\eta}\epsilon_1^2, \tag{62}$$

which implies the following holds

$$\mathbb{E}\|g_{k,t}\|^2 \leq \frac{\mathbb{E}[J(\tilde{\theta}_0)] - J(\theta^*)}{(m-1)K\eta} + \frac{\alpha}{2\eta}\epsilon_1^2. \tag{63}$$

For another part of Eq.(61), notice $\nu = k(m-1) + t$, then we have

$$\mathbb{E}\|\nabla J(\theta_{k,t}) - G_{k,t}\|^2 = \mathbb{E}\|\nabla J(\theta_\nu) - G_\nu\|^2 \tag{64}$$
$$\overset{(45)}{\leq} \mathbb{E}\left[\frac{L^2}{N_2}\sum_{i=1}^t\mathbb{E}\|\theta_{k,i+1} - \theta_{k,i}\|^2 + \mathbb{E}[\|G_{k-1,0} - \nabla J(\tilde{\theta}_{k-1})\|^2]\right]$$
$$\overset{(46)}{\leq} \mathbb{E}\left[\frac{L^2}{N_2}\sum_{i=1}^t\mathbb{E}\|\theta_{k,i+1} - \theta_{k,i}\|^2 + \frac{\alpha}{2}\epsilon_1^2\right]$$
$$\overset{(52)}{=} \mathbb{E}\left[\frac{L^2\alpha^2}{N_2}\sum_{i=1}^t\mathbb{E}\|g_{k,i}\|^2\right] + \frac{\alpha}{2}\epsilon_1^2$$
$$\overset{t\leq m}{\leq} \mathbb{E}\left[\frac{L^2\alpha^2}{N_2}\sum_{i=1}^{m-1}\mathbb{E}\|g_{k,i}\|^2\right] + \frac{\alpha}{2}\epsilon_1^2$$
$$\leq \frac{L^2\alpha^2}{KN_2}\sum_{k=1}^K\sum_{t=1}^{m-1}\mathbb{E}\|g_{k,t}\|^2 + \frac{\alpha}{2}\epsilon_1^2 \tag{65}$$
$$\overset{(59)}{\leq} \frac{L^2\alpha^2}{KN_2\eta}\left(\mathbb{E}[J(\tilde{\theta}_0)] - J(\theta^*)\right) + \left(\frac{L^2\alpha^3(m-1)}{2N_2\eta} + \frac{\alpha}{2}\right)\epsilon_1^2, \tag{66}$$

Eq.(65) holds due to the fact that the probability of selecting $\nu = k \cdot (m-1) + t$ is less than $\frac{1}{K}$.

Taking Eq(62) and Eq.(65) into Eq.(61), then we have the following inequity

$$\mathbb{E}\|\tilde{g}_{k,t}\|^2 \leq \left(\frac{1}{(m-1)K\eta} + \frac{L^2\alpha^2}{KN_2\eta\zeta^2}\right)\left(\mathbb{E}[J(\tilde{\theta}_0)] - J(\theta^*)\right) + \left(\frac{L^2\alpha^3(m-1)}{2N_2\eta\zeta^2} + \frac{\alpha}{2\zeta^2} + \frac{\alpha}{2\eta}\right)\epsilon_1^2.$$

Recall $\alpha = \frac{1}{4L}$, $N_1 = \dfrac{\left(\frac{1}{8L\zeta^2} + \frac{1}{2(\zeta - \frac{5}{32})}\left(1 + \frac{1}{32\zeta^2}\right)\right)\sigma^2}{\epsilon^2}$, $N_2 = m - 1 = $

$\dfrac{\sqrt{\left(\frac{1}{8L\zeta^2} + \frac{1}{2(\eta - \frac{5}{32})}\left(1 + \frac{1}{32\zeta^2}\right)\right)}\sigma}{\epsilon}$, then we have

$$\mathbb{E}\|\mathcal{G}_{\alpha,\langle-\nabla J(\tilde{\theta}_K),\theta\rangle}\|^2 = \mathbb{E}\|\tilde{g}_{k,t}\|^2 \leq \frac{4L}{K(m-1)(\zeta-\frac{5}{32})}\left(1 + \frac{1}{16\zeta^2}\right)(\mathbb{E}[\mathcal{J}(\tilde{\theta}_0)] - \mathcal{J}(\theta^*)) + \frac{1}{2}\epsilon^2. \tag{67}$$

Furthermore, $K = \dfrac{8L(1 + \frac{1}{16\zeta^2})}{(m-1)(\zeta - \frac{5}{32})} \cdot \dfrac{\mathbb{E}[\mathcal{J}(\tilde{\theta}_0)] - \mathcal{J}(\theta^*)}{\epsilon^2}$, we have

$$\mathbb{E}[\|\mathcal{G}^{\psi}_{\alpha,\langle-\nabla J(\tilde{\theta}_K),\theta\rangle}(\tilde{\theta}_K)\|] \leq \epsilon. \tag{68}$$

$\square$

# D    Experiments

## D.1    Experiments Details of Figure 2

For all $\ell_p$, we set $p \in [P] = \{1.1, 1.2, \cdots, 1.9, 2, 3, 4, 5\}$, we set $\gamma = 0.99$. The learning rate is chosen by grid search from the set $\{0.01, 0.02, 0.04, 0.08, 0.1\}$. For our implementation of MPO, we use a two layer feedforward neural network of 200 and 100 hidden nodes respectively, with rectified linear units (ReLU) between each layer.

## D.2    Some Practical Tricks for the Implementation of VRMPO

We present the details of the practical tricks we apply to VRMPO in the following Algorithm 3.

**(I)** For the complex real-world domains, we should tune necessitate meticulous hyper-parameter. In order to improve sample efficiency, we draw on the technique of Double Q-learning (Van Hasselt et al., 2016) to VRMPO.

**(II)** For Algorithm 2, the update rule of policy gradient (28)/(25) is a full-return update according to $R(\tau)$, which is the expensive Monte Carlo method and it tends to learn slowly. In practice, we use the one-step actor-critic structure. Let $\mathcal{D}$ be the replay memory, replacing the term $\frac{1}{N_2}\sum_{j=1}^{N_2}(-g(\tau_j|\theta_{k,t}) + g(\tau_j|\theta_{k,t-1}))$ (in (25)) as the following $\delta_{k,t}$

$$\delta_{k,t} = \frac{1}{N_2}\sum_{i=1}^{N_2}(\nabla_\theta L_{\theta_{k,t}}(s_i, a_i) - \nabla_\theta L_{\theta_{k,t-1}}(s_i, a_i)), \tag{69}$$

where $L_\theta(s, a) = -\log \pi_\theta(s, a) Q_\omega(s, a)$ is the training loss of actor, $\{(s_i, a_i)\}_{i=1}^N \sim \mathcal{D}$, $Q_\omega(s, a)$ is an estimate of action-value that can be trained to minimize the loss of critic

$$L_\omega = \frac{1}{N}\sum_{i=1}^N(Q_{\omega_{t-1}}(s_i, a_i) - Q_\omega(s_i, a_i))^2. \tag{70}$$

More details of implementation are provided in the following Algorithm 3.

In this section, we use $\ell_p$ as the mirror map.

---

**Algorithm 3** On-line VRMPO

---

**Initialize:** Policy $\pi_\theta(a|s)$ with parameter $\tilde{\theta}_0$, mirror map $\psi$,step-size $\alpha > 0$, epoch size $K$,$m$.
**Initialize:** Parameter $\tilde{\omega}_0^j, j = 1, 2$ ,$0 < \kappa < 1$ .
**for** $k = 1$ **to** $K$ **do**
  **for** each domain step **do**
    $a_t \sim \pi_{\tilde{\theta}_{k-1}}(\cdot|s_t)$
    $s_{t+1} \sim P(\cdot|s_t, a_t)$
    $\mathcal{D} = \mathcal{D} \cup \{(s_t, a_t, r_t, s_{t+1})\}$
  **end for**
  sample mini-batch $\{(s_i, a_i)\}_{i=1}^N \sim \mathcal{D}$
  $\theta_{k,0} = \tilde{\theta}_{k-1}, \omega_{k,0} = \tilde{\omega}_{k-1}^j, j = 1, 2$
  $L_\theta(s, a) = -\log \pi_\theta(s, a) \quad (\underbrace{\min_{j=1,2} Q_{\omega_{k-1}^j}(s, a)}_{\text{Double Q-Learning (Van Hasselt et al., 2016)}})$

  $\theta_{k,1} = \theta_{k,0} - \alpha_k G_{k,0}, \quad \text{where } G_{k,0} = \frac{1}{N} \sum_{i=1}^N \nabla_\theta L_\theta(s_i, a_i)\Big|_{\theta=\theta_{k,0}}$
  **for** $t = 1$ **to** $m - 1$ **do**
    /* Update Actor $(m-1)$ Epochs */
        sample mini-batch $\{(s_i, a_i)\}_{i=1}^N \sim \mathcal{D}$

$$\delta_{k,t} = \frac{1}{N} \sum_{i=1}^N \nabla_\theta L_\theta(s_i, a_i)\Big|_{\theta=\theta_{k,t}} - \frac{1}{N} \sum_{i=1}^N \nabla_\theta L_\theta(s_i, a_i)\Big|_{\theta=\theta_{k,t-1}} \tag{71}$$

$$G_{k,t} = \delta_{k,t} + G_{k,t-1} \tag{72}$$

$$\theta_{k,t+1} = \arg\min_u \{\langle G_{k,t}, u\rangle + \frac{1}{\alpha_k} D_\psi(u, \theta_{k,t})\} \tag{73}$$

  **end for**
  **for** $t = 1$ **to** $m - 1$ **do**
    /* Update Critic $(m-1)$ Epochs */
        sample mini-batch $\{(s_i, a_i)\}_{i=1}^N \sim \mathcal{D}$

$$L_{\omega_{k-1,t-1}^j}(\omega) = \frac{1}{N} \sum_{i=1}^N (Q_{\omega_{k-1,t-1}^j}(s_i, a_i) - Q_\omega(s_i, a_i))^2, j = 1, 2 \tag{74}$$

$$\omega_{k,t}^j = \arg\min_\omega L_{\omega_{k-1,t-1}^j}(\omega), j = 1, 2 \tag{75}$$

  **end for**
  $\tilde{\theta}_k \stackrel{\text{def}}{=} \theta_{k,m-1}, \tilde{\omega}_k^j \stackrel{\text{def}}{=} \omega_{k,m-1}^j, j = 1, 2$
  /* Soft Update */
  $\tilde{\theta}_k \leftarrow \kappa\tilde{\theta}_{k-1} + (1 - \kappa)\tilde{\theta}_k$
  $\tilde{\omega}_k^j \leftarrow \kappa\tilde{\omega}_{k-1}^j + (1 - \kappa)\tilde{\omega}_k^j, j = 1, 2$
**end for**

---

### D.3 TEST SCORE COMPARISON

We compare the VRMPO with baseline algorithm on test score. All the results are shown in the following Figure 6.

### D.4 MAX-RETURN COMPARISON

We compare the VRMPO with baseline algorithm on max-return. All the results are shown in the following Figure 7.

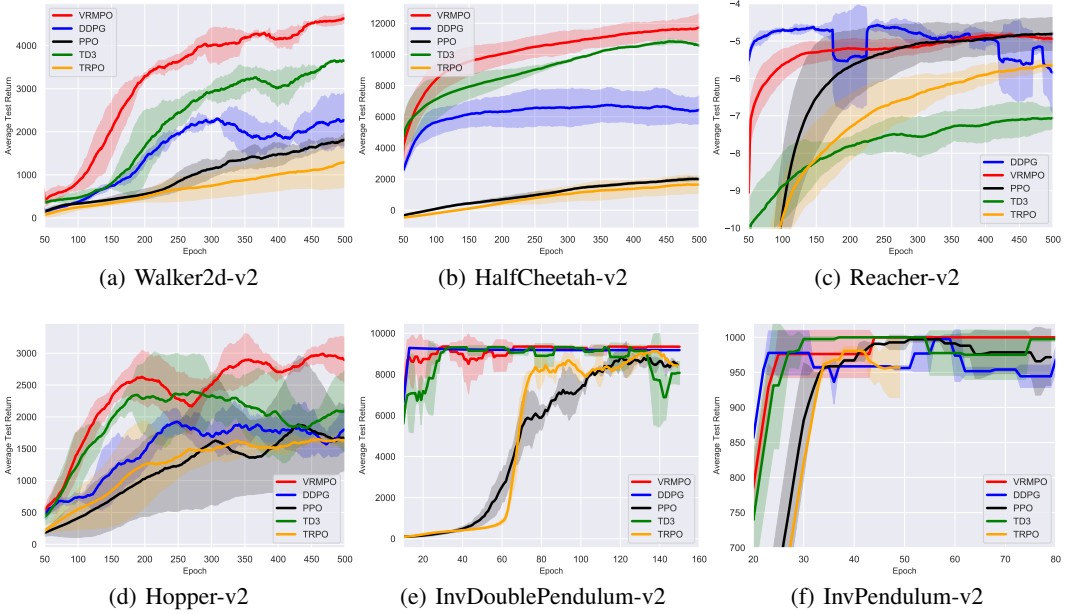

Figure 6: Learning curves of test score over epoch, where we run 5000 iterations for each epoch. The shaded region represents the standard deviation of the test score over the best 3 trials. Curves are smoothed uniformly for visual clarity.

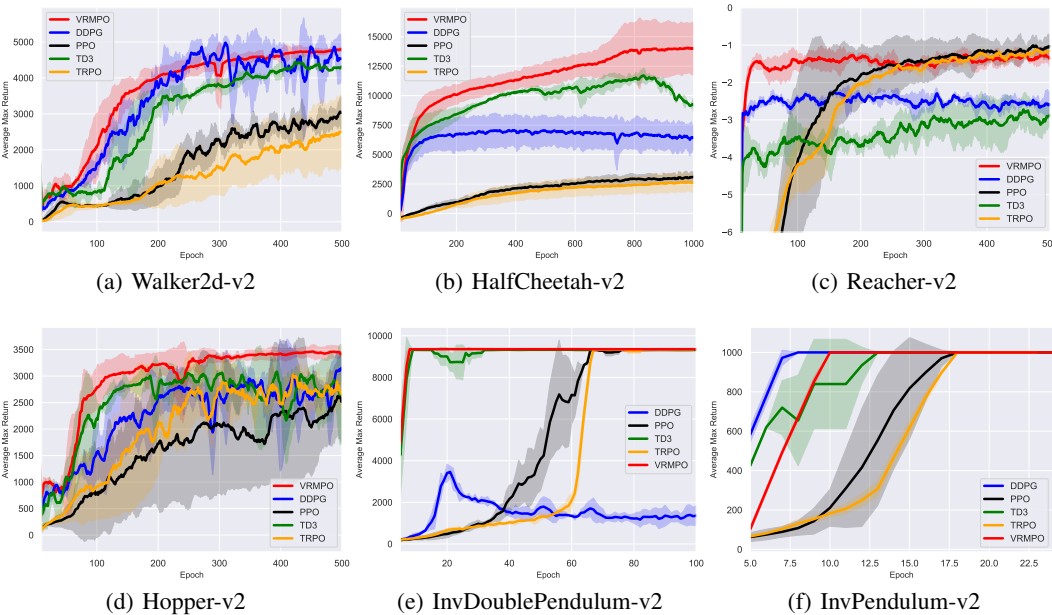

Figure 7: Learning curves of max-return over epoch, where we run 5000 iterations for each epoch. The shaded region represents the standard deviation of the test score over the best 3 trials. Curves are smoothed uniformly for visual clarity.

### D.5 DETAILS OF BASELINE IMPLEMENTATION

For all algorithms, we set $\gamma = 0.99$. For VRMPO, the learning rate is chosen by grid search from the set $\{0.1, 0.01, 0.004, 0.008\}$, batch-size $N = 100$. Memory size $|\mathcal{D}| = 10^6$. We run 5000 iterations for each epoch.

**DDPG** For our implementation of DDPG, we use a two layer feedforward neural network of 400 and 300 hidden nodes respectively, with rectified linear units (ReLU) between each layer for both the actor and critic, and a final tanh unit following the output of the actor. This implementation is largely based on the recent work by (Fujimoto et al., 2018).

**TD3** For our implementation of TD3, we refer to the work TD3 (Fujimoto et al., 2018) and https://github.com/sfujim/TD3.

We excerpt some necessary details about the implementation of TD3 (Fujimoto et al., 2018). TD3 maintains a pair of critics along with a single actor. For each time step, we update the pair of critics towards the minimum target value of actions selected by the target policy:

$$y = r + \gamma \min_{i=1,2} Q_{\theta'_i}(s', \pi_{\phi'}(s') + \epsilon),$$
$$\epsilon \sim \text{clip}(\mathcal{N}(0, \sigma), -c, c).$$

Every $d$ iterations, the policy is updated with respect to $Q_{\theta_1}$ following the deterministic policy gradient algorithm. The target policy smoothing is implemented by adding $\epsilon \sim \mathcal{N}(0, 0.2)$ to the actions chosen by the target actor network, clipped to $(-0.5, 0.5)$, delayed policy updates consists of only updating the actor and target critic network every $d$ iterations, with $d = 2$. While a larger $d$ would result in a larger benefit with respect to accumulating errors, for fair comparison, the critics are only trained once per time step, and training the actor for too few iterations would cripple learning. Both target networks are updated with $\tau = 0.005$.

**TRPO and PPO** For implementation of TRPO/PPO, we refer to https://github.com/openai/baselines/tree/master/baselines and https://spinningup.openai.com/en/latest/algorithms/trpo.html.

