# OpenReview forum: "Policy Optimization with Stochastic Mirror Descent"
_ICLR.cc/2020/Conference — Reject_

### Official Review · AnonReviewer2 · 2019-10-22
**Official Blind Review #2**

**Rating:** 3

**Review:**

This paper proposed a variant of policy gradient algorithm with mirror descent update, which is a natural generalization of projected policy gradient descent. The authors also proposed a variance reduced policy gradient algorithm following the variance reduction techniques in optimization. The authors further proved the convergence of the proposed algorithms and some experiments are conducted to show the effectiveness of their algorithms. The paper is not written in a very good way due to many typos and misleading notations. Moreover, there seem to be some technical issues in the proofs of both main theorems.

The notations are inconsistent. In eq (14) the trajectory index is k, while in eq (16) the index changed to n. In eq (28), the trajectory is denoted by $\tau_j^t$, while in the algorithm, there is no index $t$ in $\tau_j$.

Definition of P in eq (36) seems to overlap with that of $\mathcal{G}$ in eq (10), which means the same quantity was defined twice using two different notations.

Section 3.1 is almost the same as in Ghadimi et al., (2016), both the theorem and remarks. Since this is not a new finding, I suggest the authors to simplify the current statement in this subsection.

In Theorem 1 & 2, the authors used the criteria $\mathbb{E}[\|\mathcal{G}(\theta_n)\|_2^2]$ to establish the convergence result. However, they used $\mathbb{E}[\|\mathcal{G}(\theta_n)\|_2]$ to establish the convergence result in Theorem 3. This is kind of confusing because in different criteria, the complexity result will be different.

It seems that eq (32) is the same as eq (28). Can the authors elaborate the differences discussed in the paragraph after eq (32) in more detail?

In eq (40), $\hat g_k$ is defined as the average gradient over all the iterates up to $k$, which means $\hat g_k$ is a function of $\theta_1,\ldots,\theta_k$. But $g_k$, according to eq (21) in Algorithm 1, is just defined based on the current policy parameter $\theta_k$. I am skeptical why these two terms will be equal. This seems to be a technical issue of the whole proof of Theorem 2.

The derivation in (55) seems incorrect. In particular, the authors called Lemma 3, which is the result in Fang et al. (2018). However, the recursive gradient is defined differently in the current paper and in Fang’s paper. In Fang’s paper, at different iterations, the data are sampled from an unknown but fixed data distribution. But in this paper, since the data are sampled from the current policy which varies at each iteration, the data distribution is changing all the time. Therefore, Lemma 3 does not hold in the setting of this paper.

In all the experiments, it is strange that the proposed VRMPO was not compared with any of the other variance reduced algorithms listed in Table 1. I think it would be more convincing to demonstrate the improvement in Table 1 by comparing the proposed VRMPO with at least one other variance reduced algorithm.

Other comments:

1. On page 15, the first sentence of the proof: “trjecories” -> “trajectories”
2. What is $\mathcal{J^*}$ in eq (39)? It was not defined in the paper.
3. In the statement of Lemma 3, the authors said “telescoping Eq. (3) ...” However, Eq (3) in this paper is the definition of policy gradient. I assume this is a typo?
4. What is $\epsilon_1$ in eq (50)? It seems that this term does not appear in Lemma 4.

====after rebuttal====
Thanks for the authors' response. I am not satisfied with their response since they did not directly answer many of my questions about the technical flaws in this paper.

The response of the authors on the difference between estimator in Shen et al (2019) and the estimator in this paper is incorrect. The estimator in Shen et al. (2019) can also be constructed recursively based on G_0.

The authors also claimed that equation (40) is correct. However, this is not true. Although the authors removed $\hat g_k$ from equation (40), they still used the property that $\hat g_k$ is an unbiased estimator in the equation after (40). Note that $\epsilon_k$ is defined based on $\hat g_k$. Therefore, the proof is still problematic. The authors did not show how to fix it.

A more important issue of the direct use of Lemmas from Fang et al. (2018) is not fixed either. In their response, the authors simply changed the data generation in the inner loop of their algorithm such that all the iterates generate data from the initial policy $\pi_{\theta_k,0}$. However, this will cause the expectation of all the stochastic gradients used in their proof to be equal to the true policy gradient at the initial point, i.e., $\nabla J(\theta_{k,0})$. In other words, they can never obtain the full gradient $\nabla J(\theta_{k,t})$ even though their goal is to bound $\|\theta_{k,t+1}-\theta_{k,t}\|^2$ which definitely requires an approximation of $\nabla J(\theta_{k,t})$. Therefore, the issue cannot be fixed by simply using the same distribution in all the inner loop iterates. I suggest the authors to read more carefully the paper by Fang et al. (2018) and Papini et al. (2018) to understand the difference in the two settings.


**Experience Assessment:**

I have published one or two papers in this area.

**Review Assessment: Checking Correctness Of Derivations And Theory:**

I carefully checked the derivations and theory.

**Review Assessment: Checking Correctness Of Experiments:**

I assessed the sensibility of the experiments.

**Review Assessment: Thoroughness In Paper Reading:**

I read the paper thoroughly.

---

> ### Author Response · Authors · 2019-11-08
> **We have added two experiments in Appendix D.6**
>
> For your comment "In all the experiments, it is strange that the proposed VRMPO was not compared with any of the other variance reduced algorithms listed in Table 1. I think it would be more convincing to demonstrate the improvement in Table 1 by comparing the proposed VRMPO with at least one other variance reduced algorithm."
>
>
>
> Please see the revision of our submission, we have added two experiments in Appendix D.6 and the result is shown in Figure 8, which shows that the proposed VRMPO outperforms both HAPG and SVRPG in the MountainCar and CartPole domains. Besides, VRMPO achieves a better score with a lower variance.  In the final version, we will refine this section.

---

> ### Author Response · Authors · 2019-11-11
> **Reply to Paper613 AnonReviewer2**
>
> Many thanks for your very insightful comments and suggestions. We will certainly correct the typos mentioned by the reviewer. We wish to clear all your concerns as follows:
>
> (I) In Eq.(14), the mathematical symbol $0\leq k\leq N-1$ is the index of trajectory $\mathcal{T}$, while in Eq.(16), $n$ is the index of the parameters $\theta$.
>
> By Eq.(15), we generate the parameters with size $N+1$: $\{\theta_0,\theta_1,\cdots,\theta_{N}\}$. Thus, in Eq.(16), to reflect the difference of the index between parameters and trjectory $\mathcal{T}$, we introduce index $n$ to the parameters: $\{\theta_n\}_{n=0}^{N}$. In Eq.(16), we define a distribution over $\{n: n=1,2,\cdots,N\}$. While if we use $k$ (which ranges from $0$ to $N-1$) as the index, Eq.(16) is the same as $P(\tilde{\theta}_{N}=\theta_{k+1})=\frac{\zeta\alpha_{k+1}-L\alpha^{2}_{k+1}}{\sum_{i=1}^{N}(\zeta\alpha_{i}-L\alpha^{2}_{i})},
> $which is a distribution on $\{k=0,\cdots,N-1\}$.
> Thus, we think the definition of Eq.(16) is more natural.
>
> In Algorithm2, we have defined $\tau_j$ in Line 3 and Line 6. Algorithm 2 is a complete and self-consistent pseudo-code. The key idea of our VRMPO (Algorithm 2) is presented in Eq.(27)-(29).
>
>
>
> (II) The definition of $\mathcal{G}$ in Eq.(10) is the same as $P$ in Eq.(36).
>
> For Eq.(36), we use the same notation as [Ghadimi et al., 2016], and we intend to make it convenient for readers to look up the important lemma from [Ghadimi et al., 2016]. We will clear this point to improve the readability of our paper in the final version.
>
>
>
> (III) Section 3.1 is one of the contributions in our paper, especially in the open problem section, we discuss a significant problem about the convergence of policy optimization with SMD. The open problem could play a significant guiding for future work with respect to reinforcement learning theory.
>
> In section 3.1, we define the update rule of policy optimization with SMD. Then we analyze the theoretical dilemma of the convergence of the proposed method. Although Theorem 1 is a direct application of [Ghadimi et al., 2016], Theorem 1 presents a more specific analysis for the RL problem, which is a fresh understanding of policy gradient methods.
> Finally, we discuss an open problem that is proposed for the first time, i.e., this open problem never appears in any RL literature.  We all think this is a worth-pondering problem in RL community.
>
>
>
> (IV)Both Theorem 1 and Theorem 2 only present the convergence rate; they are not involved in complexity. In our paper, we don't compare the convergence rate (or complexity) of Theorem 1-2 with Theorem3 or some related works.   Thus, it's reasonable to introduce different criteria to present Theorem 1-2 and Theorem3.
>
> Theorem 3 implies the VRMPO needs $\mathcal{O}(\mathcal{\epsilon}^{-3})$ samples to achieve $\|\mathcal{G}\|<\epsilon$. Particularly, if $\psi$ is $\ell_2$-norms, the criteria $\|\mathcal{G}\|$ is reduced to gradient: $\|\nabla J\|$.
> Besides, in Table 1, we use the same criteria ($\|\nabla J\|<\epsilon$) to compare the complexity of VRMPO  with some related works, which we have presented in the caption of Table 1.
>
>
>
> (V)For Eq.(28), our VRMPO need only an initial $G_{0}$, then the gradient estimator $\{G_1,G_2,\cdots,G_n\}$ is generated according to Eq.(28).
> While for Eq.(32), for each step $t$, to get gradient estimator $G_{t}$, the HAPG needs an additional unbiased estimator $\tilde{G}_{t-1}$ to calculate $G_{t}$.According to HAPG [Shen, et,al.;2019], \[\tilde{G}_{t-1}=g(\tau_{t-1}|\theta_{t-1}),\]
> which is defined in Eq.(14).
>
>
>
>
> (VI)The reviewer's comment of Eq.(40) is correct.
> In Eq.(40), we actually want to express "$\mathbb{E}[g_k]={\nabla J(\theta_k)}$", without the term $\hat{g}_{k}$.
>
> Fortunately, with a double-check, we find the rest proof of Eq.(41)-(42) is correct, which is not affected by the term $\hat{g}_{k}$. We will correct this mistake in the final version.
> We are very sorry due to our careless operation misled the reviewer. Please rethinking our proof.
>
>
>
> (VII)For your comment about Eq.(55) and Fang’s paper.
> In fact, for the line 6 of Algorithm 2 table, we actually want to present" $\{\tau_j\}_{j=1}^{N2}\sim \pi_{\theta_{k,0}}$" not $\pi_{\theta_{k,t}}$, i.e., in the inner loop, we sample the data according to the policy determined by the outer loop. Thus, for the inner loop, the data is generated by a fixed distribution.
>
> In practice, for the inner loop, we sample from a fixed distribution, please see Page24, Algorithm 3, Eq.(71)-Eq.(73) and Eq.(74)-(75).
>
> We will refine this point in the final version. We are all very sorry again due to our careless operation misled the reviewer. Please rethinking our proof.
>
>
>
>
> (VIII) Other problems.
>
> $\mathcal{J}^{*}$ is short for $J(\theta^*)$.
>
> We donate the complex term in (50) as $\epsilon_1^{2}$ to simplify symbol, i.e., the term of Eq.(50) is $\epsilon_1^{2}$.

---

### Official Review · AnonReviewer3 · 2019-10-23
**Official Blind Review #3**

**Rating:** 3

**Review:**


[Summary]
This paper proposes MPO, a policy optimization method with convergence guarantees based on stochastic mirror descent that uses the average of previous gradients to update the policy parameters. A lower-variance method, VRMPO, is then proposed that matches the best known convergence rate in the in the literature. Experiments show that (1) MPO converges faster than basic policy optimization methods on a small task, and (2) VRMPO achieves a performance comparable to, and often better than, popular policy optimization methods (TD3, DDPG, PPO, and TRPO) on MuJoCo.

[Decision]
The proposed methods are well-grounded. The experiments are, however, limited and miss important baselines discussed in previous sections. The presentation is not clear. Imprecise statements, undefined terms, and grammatical errors make the paper hard to follow. Overall, I am leaning towards rejecting the paper.

[Explanation]
Section 5 provides a comparison between the convergence rates of VRMPO and previous methods (VPG, REINFORCE, SVRPG, and HAPG). I was expecting to see a similar comparison in the experiments section but SVRPG and HAPG do not appear in the experiments although their convergence rates are comparable to VRMPO. I especially want to know how VRMPO differs from HAPG. Their convergence rates are the same and their updates look similar.

The description below Fig. 1 says that VRMPO converges faster and achieves a better performance than the baselines. While VRMPO does converge faster, all the three methods seem to converge to the same (optimal) solution.

The paper needs more polished presentation. Here are some examples for improving the writing:
- The introduction section: "our algorithm outperforms state-of-the-art bandit algorithms in...." The compared methods are RL algorithms.
- "converges faster significantly and achieves a better performance than both REINFORCE and MPO" -> "... than both REINFORCE and VPG".
- "The task is to estimate the value function of state s_1" -> This seems to be a control task where the goal is to achieve the highest return.
- "Eq.(23) has a closed implementation" -> What is a closed implementation?
- Some terms like "projected gradient" and "baseline" (in the context of variance reduction) are not defined. The compared methods in 6.2 are not described in the paper (except TD3 whose brief description appears in D.5).
- "The traditional policy gradient methods such as REINFORCE, VPG, and DPG are all the algorithms update parameters in Euclidean distance" -> "... such as REINFORCE, VPG, and DPG update parameters in Euclidean distance"
- In Table 1 the performance of VRMPO on Reacher is in bold while it is not the maximum value.
- "It is different from (Du et al., 2017)..." -> "VRMPO is different from..."
- "due to it doesn't require..." -> "because it does not require..."

[Minor comments]
- How is the bound in Eq. (18) proved?


**Experience Assessment:**

I have published one or two papers in this area.

**Review Assessment: Checking Correctness Of Derivations And Theory:**

I assessed the sensibility of the derivations and theory.

**Review Assessment: Checking Correctness Of Experiments:**

I carefully checked the experiments.

**Review Assessment: Thoroughness In Paper Reading:**

I read the paper at least twice and used my best judgement in assessing the paper.

---

> ### Author Response · Authors · 2019-11-09
> **Reply to Paper613 AnonReviewer3**
>
> We wish to clear all your concerns as follows:
>
>
> (I)For your comment, "I especially want to know how VRMPO differs from HAPG. Their convergence rates are the same and their updates look similar."
>
> For our VRMPO, i.e., the update rule (28) that needs only an initial $G_{0}$, then the gradient estimator $\{G_1,G_2,\cdots,G_n\}$ is generated according to Eq.(28). While for Eq.(32), for each time $t$, to get gradient estimator $G_{t}$, the HAPG needs an additional unbiased estimator $\tilde{G}_{t-1}$ to calculate $G_{t}$. According to HAPG [Shen, et,al.;2019], \[\tilde{G}_{t-1}=g(\tau_{t-1}|\theta_{t-1}),\]
> which is defined in Eq.(14).
>
> For your comment “I was expecting to see a similar comparison in the experiments section but SVRPG and HAPG do not appear in the experiments although their convergence rates are comparable to VRMPO.”
>
> Please see the revision of our submission, we have added two experiments in Appendix D.6 and the result is shown in Figure 8, which shows that the proposed VRMPO outperforms both HAPG and SVRPG in the MountainCar and CartPole domains. Besides, VRMPO achieves a better score with a lower variance.  In the final version, we will refine this section.
>
> ******
> (II)For your comment, "The description below Fig.1 says that VRMPO converges faster and achieves a better performance than the baselines. While VRMPO ......to converge to the same (optimal) solution."
>
> In Fig.1,  Both VPG and REINFORCE converge nearly the value function of state $s_1$ (we denote it as $G_0$), but never reach the $G_0$. Thus, we claim "VRMPO converges faster and achieves a better performance than the baselines".
>
> ******
> (III)For your comment, "The task is to estimate the value function of state $s_1$" -> This seems to be a control task where the goal is to achieve the highest return."
>
> The task is to estimate the value function of state $s_1$: $G_0\approx-11.6$. All the details are presented on page 18.
>
> ******
> (IV)For your comment " "Eq.(23) has a closed implementation" -> What is a closed implementation?"
>
> The term "a closed implementation" means the Eq.(23) can be represented as a specific calculation form. In our experiment, we use $\ell_p$-norms as the mirror map $\psi$, our satement "Eq.(23) has a closed implementation" means Eq(23)
> \[
> \theta_{k+1}=\arg\min_{\omega}\{\langle -\hat{g}_k,\omega\rangle+\frac{1}{\alpha_k}D_{\ell_p}(\omega,\theta_k)\}
> \]
> can be implemented via the following calculation form:
> \[
> \theta_{k+1}=\nabla\psi^{*}(\nabla\psi(\theta_{k})+\alpha_{k}\hat{g}_{k}),
> \]
> where $\nabla\psi_{j}(x)$ and $\nabla\psi_{j}^{*}(y)$ is defined as:
> $
> \nabla\psi_{j}(x)=\frac{\text{sign}(x_{j})|x_j|^{p-1}}{\|x\|_{p}^{p-2}},
> \nabla\psi_{j}^{*}(y)=\frac{\text{sign}(y_{j})|y_j|^{q-1}}{\|y\|_{q}^{q-2}}, \frac{1}{p} +\frac{1}{q}=1
> $
> and $j$ is coordinate index of the vector $\nabla\psi$, $\nabla\psi^{*}$.
> We have presented all the details on Page 9.
>
> ******
> (V)For your comment "Some terms like "projected gradient" and "baseline" (in the context of variance reduction) are not defined. The compared methods in 6.2 are not described in the paper (except TD3 whose brief description appears in D.5)."
>
> (i)We will illustrate the term "projected gradient" in the final version.
>
> (ii)Your comment for the term "baseline" is related to our statement " Baseline (also known as control variates (Cheng et al., 2019a)..." on page 7?      The term "Baseline" is a widely used technique in policy gradient community, i.e., the term "Baseline"  is an algorithm.  Besides, we have cited (Cheng et al., 2019a) to guide the readers to understand this term.
>
> ******
> (VI)For your comment, "In Table 1 the performance of VRMPO on Reacher is in bold while it is not the maximum value."
>
> Your comment is correct.
> Our VRMPO archives the score "-0.49" that is very near the maximum value "-0.44".
> Besides, the score of our VRMPO outperforms significantly than the score "-1.47", "-1.55", and "-0.66" in the Reacher-v2 domain. In the final version, we will refine this section.
>
> ******
> (VII ) "How is the bound in Eq. (18) proved?"
>
> \[
> \dfrac{\big(J(\theta^{*})-J(\theta_1)\big){+\frac{\sigma^{2}}{\zeta}\sum_{k=1}^{N}{\alpha_k}}}{{\sum_{k=1}^{N}(\zeta\alpha_{k}-L\alpha^{2}_{k})}}
> \ge
> \dfrac{{\frac{\sigma^{2}}{\zeta}\sum_{k=1}^{N}{\alpha_k}}}{{\sum_{k=1}^{N}(\zeta\alpha_{k}-L\alpha^{2}_{k})}}
> \ge\dfrac{{\frac{\sigma^{2}}{\zeta}\sum_{k=1}^{N}{\alpha_k}}}{{\sum_{k=1}^{N}\zeta\alpha_{k}}}
> =\frac{\sigma^{2}}{\zeta^{2}},
> \]
> where the first $\ge$ holds because $J(\theta^{*})-J(\theta_1)\ge0$, the second $\ge$ holds because $\zeta\alpha_{k}-L\alpha^{2}_{k}\leq\zeta\alpha_{k}$

---

### Official Review · AnonReviewer1 · 2019-10-25
**Official Blind Review #2**

**Rating:** 6

**Review:**

This paper proposes a new policy gradient method that is based on stochastic mirror descent. It has been shown theoretically and empirically that the method achieves more sample efficiency.

Overall, the proposed idea is interesting. The paper has good contributions in both theoretical and algorithmic aspects to policy optimization family. The empirical results are also very promising. I have only some following concerns.

- It would be really nice if the discussions on connections to existing policy gradient methods on page 4 can be elaborated. This is to make the readers understand better how the reduction from the proposed method to an existing one can be made. It would also be more interesting after such reductions can be made, one can compare the sample efficiency of the proposed method with such state-of-the-art policy gradient methods. That means Table 1 can be extended with rows of such methods.

- The convergence analysis in Theorem 2 and 3 are really strong, but it would be very useful if there are ablations that can reflect the trade-off between the convergence property and hyperparamters used in those theorems.

- The contribution of the Bregman distance D to the update should be worth further discussions. It is an important part of SMD, but for policy optimization one might want to see how it contributes to stabilizing the update OR policy exploration. As seen, with a different choice of D, the method is reduced to policy gradient or entropy-regularized policy optimization. Ablations without D might show some interesting results too.


- Some minor comments:
page 8: "The result in Figure 1 shows that MPO converges faster significantly and achieves a better performance than both REINFORCE and MPO." -> should be "... both REINFORCE and VPG."

**Experience Assessment:**

I have read many papers in this area.

**Review Assessment: Checking Correctness Of Derivations And Theory:**

I assessed the sensibility of the derivations and theory.

**Review Assessment: Checking Correctness Of Experiments:**

I assessed the sensibility of the experiments.

**Review Assessment: Thoroughness In Paper Reading:**

I read the paper at least twice and used my best judgement in assessing the paper.

---

> ### Author Response · Authors · 2019-11-09
> **Reply to Paper613 AnonReviewer1**
>
> Many thanks for your great support and very insightful comments. We will certainly correct the typos mentioned by the reviewer. We wish to clear all your concerns as follows:
>
> (I)If $\psi=\ell_2$-norms, $D_{\psi}(x,y)=\|x-y\|_{2}^{2}$, then from the update rule (15), we have
> \[
> \theta_{k+1}=
> \arg\min_{\theta}\Big\{\langle-g_{k},\theta\rangle+\frac{1}{\alpha_k}D_{\psi}(\theta,\theta_k)\Big\}=
> \arg\min_{\theta}\Big\{\langle-g_{k},\theta\rangle+\frac{1}{\alpha_k}\|\theta-\theta_{k}\|_{2}^{2}\Big\}=\theta_k+\alpha_k g_k,
> \]
> where $g_{k}$ is short for $g(\tau_k|\theta_k)=\sum_{t\ge0}\nabla \log \pi_{\theta}R (\tau_k)$ defined in Eq.(14).
> Since then the update (15) reduces to $VPG/REINFORCE$.
>
>
> If $\psi(\theta)=\frac{1}{2}\theta^{\top}F\theta$, where $F$ is the Fisher information matrix of the distribution $\pi_{
> \theta}$, i.e.,
> \[
> F=\mathbb{E}_{\tau\sim\pi_{\theta}}[\nabla \log \pi_{\theta}\nabla^{\top} \log \pi_{\theta}],
> \]
> Thus, $D_{\psi}(\theta_1,\theta_2)=(\theta_1-\theta_2)^{\top}F(\theta_1-\theta_2)=\psi(\theta_1-\theta_2)$.
> Then, by the update rule (15), we have
> \[
> \theta_{k+1}
> =\arg\min_{\theta}\Big\{\langle-g_{k},\theta\rangle+\frac{1}{\alpha_k}D_{\psi}(\theta,\theta_k)\Big\}=\theta_k+\alpha_k F^{-1} g_k,
> \]
> which is $Natural~Policy~Gradient$ algorithm [Sham Kakade; NIPS2002].
> [Sham Kakade; NIPS2002] A Natural~Policy~Gradient.
>
>
> If $\psi(\theta)=\sum_{i=1}^{p}\theta^{i}\log \theta^i$ is Boltzmann-Shannon entropy function, where $\theta=(\theta^1,\theta^2,\cdots,\theta^{p})^{\top}$ s.t $\sum_{i}\theta^i =1$ and $\theta^i\ge0$. Usually, we set $0\log0 =0$. Thus, $D_{\psi}(\theta,\omega)=\sum_{i=1}^{p}\theta^i\log\frac{\theta^i}{\omega^i}$ is KL-Divergence.
> Then, the update rule (15) is reduce to $Relative~Entropy~Policy~Search$ [Jan Peters,et,al. AAAI2010;Roy Fox,et,al.,UAI2015]:
> \[
> [\theta_{k+1}]_i=\frac{[\theta_k]_i e^{-\alpha_k[g_k]_{i}}}{\sum_{j=1}^{p}[\theta_k]_j e^{-\alpha_k[g_k]_j}},
> \]
> where $[\theta_{k+1}]_i$ is the $i$-th coordinate component of the vector $\theta_k$, $[g_k]_i$ the $i$-th coordinate component of the vector $g_k$.
>
> [Jan Peters,et,al. AAAI2010] Relative Entropy Policy Search.
> [Roy Fox, et, al., UAI2015]Taming the Noise in Reinforcement Learning via Soft Updates.
>
>
>
> *************
> (II)For your comment and suggestion, "It would also be more interesting after such reductions can be made, one can compare the sample efficiency of the proposed method with such state-of-the-art policy gradient methods. That means Table 1 can be extended with rows of such methods."
>
>
> It is a very nice suggestion to extended with rows of Table 1. We're also trying to extend the complexity of Table 1.
> To our best knowledge, Table 1 has summarized all the related works so far. The results have appeared in Table 1 are the complexity of the fundamental versions in policy optimization family. As we known, although some simple and special versions (e.g., linear policy of natural policy gradient [Harshat Kumar, et, al., 2019] or softmax policy [Alekh Agarwal, et, al.,2019]) of the complexity have also been proposed, the results in our Table 1 are the general  and fundamental versions of policy optimization. We will clear this point in the final version.
>
>
> [Harshat Kumar, et,al., 2019] https://arxiv.org/abs/1910.08412
> [Alekh Agarwal,et,al.,2019] https://arxiv.org/abs/1908.00261
>
>
> *************
> (III)For the comment "The convergence analysis in Theorem 2 and 3 are really strong, but it would be very useful if there are ablations that can reflect the trade-off between the convergence property and hyper-paramters used in those theorems."
>
> Thank you for your insightful suggestion. In the final version, we will clear the following points:
> (i)For Theorem 2 implies the convergence result \[\|\mathcal{G}\|\leq \mathcal{O}(\ln N/ N),\]
> is determined by the following hyper-paramter: $\zeta$, the strongly convex hyper-paramter of mirror map; the Lipschiz constant $L$.
>
> (ii)Theorem 3 implies that to achieve $\|\mathcal{G}\|\leq \epsilon$, VRMPO needs $\mathcal{O}(\epsilon^{-3})$ random trajetories which contains: the iteration times of outer loop $K=\mathcal{O}(\epsilon^{-2})$,  the batch size of the trajectories of outer loop \[N_1= (m-1)^2=(N_2)^{2}=\mathcal{O}(\epsilon^{-2}),\]
> which illustrates the precise quantitative relationship between the batch size of trajectories of outer loop $N_1$, the iteration times of the inner loop $(m-1)$ and the batch size of trajectories of inner loop $N_2$.
>  The relationship of the  hyperparamters $N_1= (m-1)^2=(N_2)^{2}=\mathcal{O}(\epsilon^{-2})$ plays a critical role in achieving $\|\mathcal{G}\|\leq \epsilon$, for more details, please see our Lemma 4 in Appendix C.

---

### Author Response · Authors · 2019-11-11
**Abut the revision of Paper613**

In the revision of our submission:

 (i)we have added two experiments in Appendix D.6.

(ii)we have refined some typos according to the reviewers.

---

### Decision · Program_Chairs · 2019-12-19

**Decision:**

Reject

**Comment:**

This paper proposes a new policy gradient method based on stochastic mirror descent and variance reduction. Both theoretical analysis and experiments are provided to demonstrate the sample efficiency of the proposed algorithm. The main concerns of this paper include: (1) unclear presentation in both the main results and the proof; and (2) missing baselines (e.g., HAPG) in the experiments. This paper has been carefully discussed but even after author response and reviewer discussion, it does not gather sufficient support.

Note: the authors disclosed their identity by adding the author names in the revision during the author response. After discussion with PC chair, the openreview team helped remove that revision during the reviewer discussion to avoid desk reject.